# TBX3 acts as tissue-specific component of the Wnt/β-catenin transcriptional complex

**Dario Zimmerli[1†‡], Costanza Borrelli[2†], Amaia Jauregi-Miguel[3,4†], Simon Söderholm[3,4], Salome Brütsch[1], Nikolaos Doumpas[1], Jan Reichmuth[1], Fabienne Murphy-Seiler[5], MIchel Aguet[5], Konrad Basler[1]\*, Andreas E Moor[2]\*, Claudio Cantù[3,4]\***

[1]Department of Molecular Life Sciences, University of Zurich, Zürich, Switzerland; [2]Department of Biosystems Science and Engineering, ETH Zürich, Basel, Switzerland; [3]Wallenberg Centre for Molecular Medicine, Linköping University, Linköping, Sweden; [4]Department of Biomedical and Clinical Sciences, Division of Molecular Medicine and Virology; Faculty of Medicine and Health Sciences; Linköping University, Linköping, Sweden; [5]Swiss Institute for Experimental Cancer Research (ISREC), Ecole Polytechnique Fe´de´rale de Lausanne (EPFL), School of Life Sciences, Lausanne, Switzerland

**\*For correspondence:**
konrad.basler@imls.uzh.ch (KB);
andreas.moor@bsse.ethz.ch (AEM);
claudio.cantu@liu.se (CC)

[†]These authors contributed equally to this work

**Present address:** [‡]Division of Molecular Pathology, The Netherlands Cancer Institute, Amsterdam, Netherlands

**Competing interests:** The authors declare that no competing interests exist.

**Abstract** BCL9 and PYGO are β-catenin cofactors that enhance the transcription of Wnt target genes. They have been proposed as therapeutic targets to diminish Wnt signaling output in intestinal malignancies. Here we find that, in colorectal cancer cells and in developing mouse forelimbs, BCL9 proteins sustain the action of β-catenin in a largely PYGO-independent manner. Our genetic analyses implied that BCL9 necessitates other interaction partners in mediating its transcriptional output. We identified the transcription factor TBX3 as a candidate tissue-specific member of the β-catenin transcriptional complex. In developing forelimbs, both TBX3 and BCL9 occupy a large number of Wnt-responsive regulatory elements, genome-wide. Moreover, mutations in *Bcl9* affect the expression of TBX3 targets in vivo, and modulation of TBX3 abundance impacts on Wnt target genes transcription in a β-catenin- and TCF/LEF-dependent manner. Finally, TBX3 overexpression exacerbates the metastatic potential of Wnt-dependent human colorectal cancer cells. Our work implicates TBX3 as context-dependent component of the Wnt/β-catenin-dependent transcriptional complex.

## Introduction

The Wnt pathway is an evolutionarily conserved cell signaling cascade that acts as major driving force of several developmental processes, as well as for the maintenance of the stem cell populations within adult tissues (*Nusse and Clevers, 2017*). Deregulation of this signaling pathway results in a spectrum of consequences, ranging from lethal developmental abnormalities to several forms of aggressive cancer (*Nusse and Clevers, 2017*). Most prominently, colorectal cancer (CRC) is initiated by genetic mutations that constitutively activate Wnt signaling (*Kahn, 2014*).

Secreted WNT ligands trigger an intracellular biochemical cascade in the receiving cells that culminates in the calibrated expression of target genes (*Mosimann et al., 2009*). This transcriptional response is orchestrated by nuclear β-catenin, that acts as a 'scaffold' to buttress a host of co-factors to cis-regulatory elements occupied by the TCF/LEF transcription factors (*Valenta et al., 2012*). Among the co-factors, the two paralogs BCL9 and BCL9L (referred to as BCL9/9L) and PYGO1/2 proteins reside within the Wnt/β-catenin transcriptional complex, and their concerted action is required to efficiently activate Wnt-target gene expression (*Kramps et al., 2002*; *Parker et al.,*

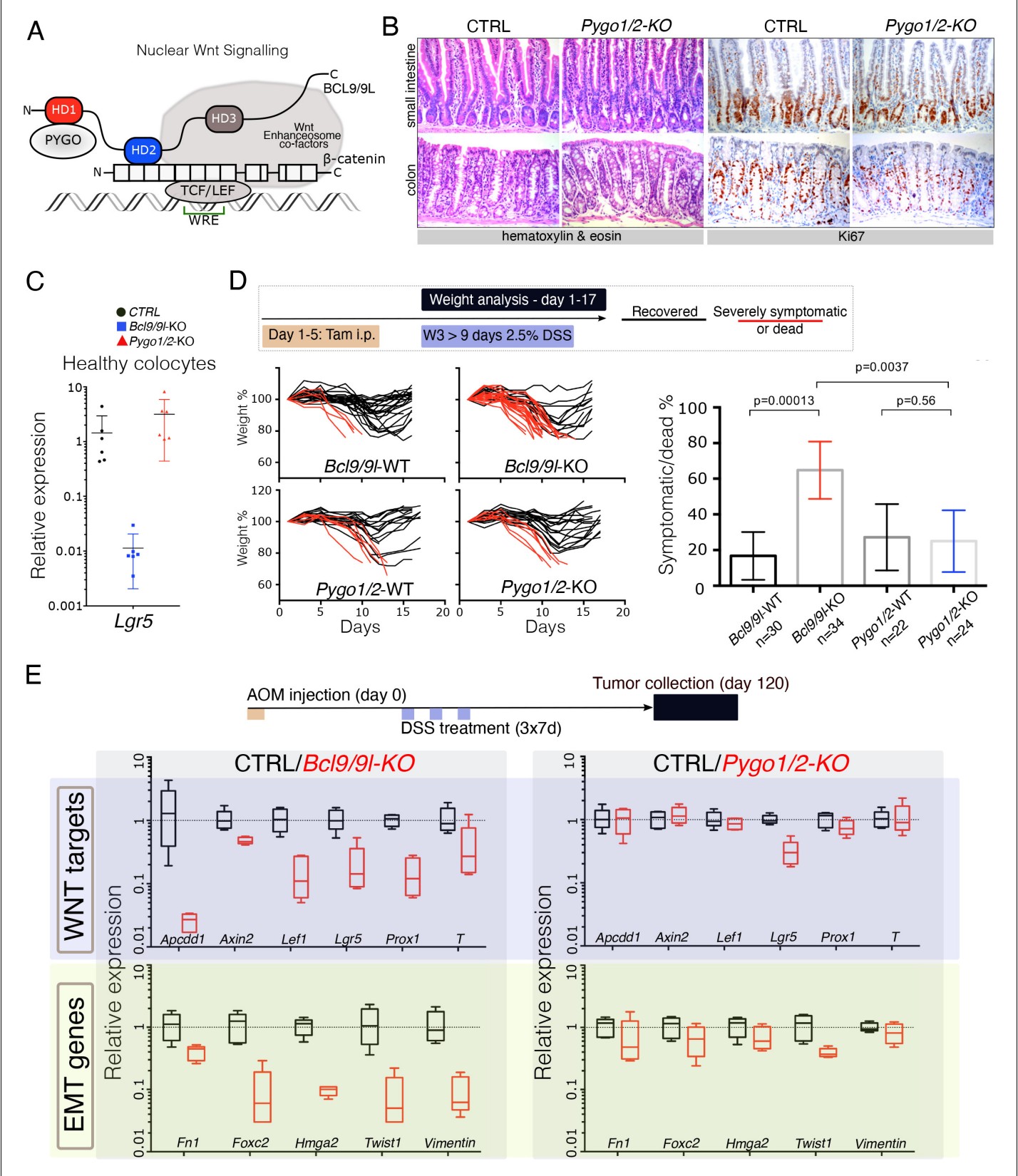

**Figure 1.** The intestinal epithelium-specific recombination of *Pygo1/2* does not recapitulate the effects of deleting *Bcl9/9l*. (**A**) Schematic representation of the Wnt/β-catenin transcriptional complex, with emphasis on the so-called 'chain of adaptors' components, β-catenin, BCL9/9L and PYGO; Wnt

*Figure 1 continued on next page*

*Figure 1 continued*

Responsive Element (WRE). The homology domains 1–3 (HD1-3) of BCL9/9L are shown. (B) Epithelial-specific *Pygo1/2* deletion (via *vil-Cre-ER^t2*; *Pygo1/2*-KO) does not lead to any obvious histological or functional defect, neither in the small intestine nor in the colon as seen by hematoxylin and eosin staining (left panels). The proliferative compartment, detected via Ki67 (right panels), seems also unaffected (also refer to the count in *Figure 1—figure supplement 1D–E*). (C) Quantitative RT-PCR detecting *Lgr5* mRNA extracted from colonic epithelium of control (black), *Bcl9/9l* (blue) or *Pygo1/2* (red) conditional mutants (KO). (D) 6–8 week-old male mice were treated with five tamoxifen (Tam) injections (i.p., 1 mg/day) for five consecutive days. 10 days later mice were treated with 2.5% dextran sodium sulfate (DSS) ad libitum in drinking water for 9 days. While 17% of control mice (N = 30) were severely affected or died due to the DSS treatment (red lines), 65% of conditional *Bcl9/9l*-KO (N = 34) mice performed poorly in this test. Deletion of *Bcl9/9l* increased significantly the death rate after DSS treatment (p-value=0.00013 in Fisher's Exact Test). No difference between *Pygo1/2*-KO and control mice could be measured: 27% of control mice (N = 22) and 25% of *Pygo1/2*-KO (N = 24) were affected upon DSS treatment (p-value=0.5626 in Fisher's Exact Test). (E) 6–8 week-old female mice were exposed to a single dose of the carcinogenic agent azoxymethane (AOM), followed by 7 days of DSS administration in the drinking water. This regimen results in the emergence of dysplastic adenomas that are collected for RNA extraction and analysis of the indicated targets via RT-PCR: Wnt target genes and genes expressed during epithelial-to-mesenchymal transition (EMT), associated with cancer metastasis.

The online version of this article includes the following figure supplement(s) for figure 1:

**Figure supplement 1.** Efficient epithelial-specific *Pygo1/2* deletion does not lead to obvious defects.
**Figure supplement 2.** Intestinal epithelium-specific recombination of *Pygo1/2* does not recapitulate the effects of deleting *Bcl9/9l*.

*2002*; *van Tienen et al., 2017*; *Figure 1A*). During vertebrate development, their requirement in the β-catenin-mediated transcription appears to be context-dependent (*Cantù et al., 2018*; *Li et al., 2007*), and they also have evolved β-catenin-independent functions (*Cantù et al., 2017*; *Cantù et al., 2014*). Curiously however, BCL9 and PYGO always seem to act as a 'duet' (*Kennedy et al., 2010*).

Importantly, BCL9/9L and PYGO proteins were found to significantly contribute to the malignant traits typical of Wnt-induced CRCs (*Deka et al., 2010*; *Gay et al., 2019*; *Jiang et al., 2020*; *Mani et al., 2009*; *Mieszczanek et al., 2019*; *Moor et al., 2015*; *Talla and Brembeck, 2016*). These observations provided impetus to consider the BCL9/PYGO axis as relevant 'targetable' unit in CRC (*Lyou et al., 2017*; *Mieszczanek et al., 2019*; *Talla and Brembeck, 2016*; *Zimmerli et al., 2017*).

However, here we noticed an apparent divergence between the roles of BCL9/9L and PYGO proteins. We found that genetic abrogation of *Bcl9/9l* in mouse CRC cells results in broader consequences than *Pygo1/2* deletion, suggesting that BCL9 function does not entirely depend on PYGO1/2. Among the putative β-catenin/BCL9 interactors we identified the developmental transcription factor TBX3. Intriguingly, we show that also during forelimb development, BCL9/9L possess a PYGO-independent role. In this in vivo context, TBX3 occupies β-catenin/BCL9 target loci genome-wide, and mutations in *Bcl9/9l* affect the expression of TBX3 targets. Finally, TBX3 modulates the expression of Wnt target genes in a β-catenin- and TCF/LEF-dependent manner, and increases the metastatic potential of human CRC cells when overexpressed. We conclude that TBX3 can assist the Wnt/β-catenin mediated transcription in selected developmental contexts, and that this partnership could be aberrantly reactivated in some forms of Wnt-driven CRCs.

## Results and discussion

We induced intestinal epithelium-specific recombination of *Pygo1/2* loxP alleles (*Pygo1/2*-KO), that efficiently deleted these genes in the whole epithelium, including the stem cells compartment (*Figure 1—figure supplement 1A and B*). Consistently with recent reports (*Mieszczanek et al., 2019*; *Talla and Brembeck, 2016*), and similarly to deletion of *Bcl9/9l* (*Deka et al., 2010*; *Mani et al., 2009*; *Moor et al., 2015*), *Pygo1/2*-KO displayed no overt phenotypic defects (*Figure 1B*; *Figure 1—figure supplement 1C–E*). We were surprised in noticing that the expression of *Lgr5*, the most important intestinal stem cell marker and Wnt target gene (*Barker et al., 2007*), was heavily downregulated upon loss of *Bcl9/9l* but unaffected in *Pygo1/2*-KO (*Figure 1C*). To address the functionality of the stem cell compartment in these two conditions, we subjected both *Bcl9/9l* and *Pygo1/2* compound mutants (KO) to a model of intestinal regeneration by DSS treatment (*Kim et al., 2012*; *Figure 1D*). While *Bcl9/9l*-KO mice showed a defect in regeneration after insult (*Deka et al., 2010*), *Pygo1/2*-KO proved indifferent when compared to control littermates (*Figure 1D*). While we cannot exclude that PYGO1/2 also contributes to the Wnt/β-catenin-

dependent transcriptional regulation, our results highlight that the BCL9/9L function in the intestinal epithelium homeostasis and regeneration does not entirely depend on PYGO1/2. This was surprising, since BCL9/9L proteins were thought to act as mere 'bridge' proteins that tethered PYGO to the β-catenin transcriptional complex (*Figure 1A*; *Fiedler et al., 2015*; *Mosimann et al., 2009*). Both BCL9 and PYGO proteins have been implicated in colorectal carcinogenesis (*Gay et al., 2019*; *Jiang et al., 2020*; *Mieszczanek et al., 2019*; *Talla and Brembeck, 2016*). We tested if the consequence of the deletion of *Bcl9/9l* and *Pygo1/2* genes was also different in the context of carcinogenesis. Specifically, we looked at the contribution to gene expression in chemically-induced AOM/DSS colorectal tumors (*Figure 1E*). As previously observed, *Bcl9/9l*-KO tumors exhibit a massive decrease in Wnt target gene expression, epithelial-to-mesenchymal transition (EMT) and stemness traits (*Deka et al., 2010*; *Moor et al., 2015*), which was not observed in *Pygo1/2*-KO tumors (*Figure 1E*, *Figure 1—figure supplement 2A and B*). This phenotypic difference is consistent with a recent study in which *Bcl9/9l* but not *Pygo1/2* loss reduced the activation of Wnt target genes induced by *APC* loss-of-function (*Mieszczanek et al., 2019*); we interpret this as an independent validation of our observation. All these experiments open up the question of how BCL9/9L imposes its function independently of PYGO.

Surprisingly, the intestine-specific deletion of the homology domain 1 (HD1) of BCL9/9L (*Figure 2A*), that was previously annotated to interact only with PYGO1/2 (*Cantù et al., 2014*; *Kramps et al., 2002*), i) suppressed the metastatic phenotype of the AOM/DSS tumors while deletion of *Pygo1/2* did not (*Figure 2B*) and ii) induced a strong downregulation of Wnt target, EMT and stemness genes (*Figure 2C*, *Figure 2—figure supplement 1*). The discrepancy between the gene expression changes induced by recombining *Pygo1/2* or deleting the HD1 domain of *Bcl9/9l* implies that currently unknown proteins assist BCL9/9L function. We set out to identify new candidate BCL9 partners that might be responsible for the different phenotypes. To this aim, we performed a pull-down of tumor proteins expressing either a full-length or a HD1-deleted variant of BCL9, followed by mass spectrometry (*Figure 2D*). Among the proteins differentially pulled down by control but not by mutant BCL9 we detected TBX3 (*Figure 2D and E*) and selected it for further validation. The in vivo deletion of the HD1 domain (in *Bcl9/9l-ΔHD1* embryos) leads to severe forelimb malformations, while *Pygo1/2*-KO embryonic forelimbs are unaffected (*Figure 2F*)(also see *Schwab et al., 2007*). Limb development, thus, represents another context where BCL9/9L appear to act independently of PYGO. Of note, TBX3 plays a fundamental role in the development of this structure (*Frank et al., 2013*).

We confirmed cytological vicinity between transfected tagged versions of BCL9 and TBX3 by proximity ligation assay (PLA) (*Figure 2—figure supplement 2A,B*). However, overexpression-based in vitro co-immunoprecipitation experiments could not detect any stable interaction between these two proteins, suggesting absence of direct binding or a significantly lower affinity than that between BCL9 and PYGO (*Figure 2—figure supplement 2C*). Hence, we aimed at testing the functional association between TBX3 and BCL9 in a more relevant in vivo context. To this aim, we collected ca. 500 forelimbs from 10.5 dpc wild-type mouse embryos and subjected the crosslinked chromatin to immunoprecipitation using antibodies against BCL9 (*Salazar et al., 2019*) or TBX3, followed by deep-sequencing of the purified DNA (ChIP-seq, *Figure 3A*). By using stringent statistical parameters, and filtering with Irreproducible Discovery Rate (IDR), we extracted a list of high confidence BCL9 and TBX3 peaks (*Figure 3B and C*). Surprisingly, we discovered that BCL9 occupies a large fraction (ca. 2/3$^{rd}$) of the TBX3-bound regions (*Figure 3D*). Suggestive of a role for TBX3 within the Wnt-dependent transcriptional apparatus, motif analysis of the common TBX3-BCL9 target loci identified statistical prevalence for TCF/LEF and Homeobox transcription factor consensus sequences, but not for any TBX transcription factor (*Figure 3E*). This suggests that TBX3 interacts with the DNA in these locations via affinity to the Wnt/β-catenin co-factors rather than via direct contact with DNA. Accordingly, TBX-specific motifs were detected within the group of TBX3 exclusive peaks (which do not display BCL9 binding, *Figure 3—figure supplement 1*). Notably, TBX3 and BCL9 occupancy was detected at virtually all previously described Wnt-responsive-elements (WRE) within known Wnt target genes (*Figure 3F*).

To test whether the in vivo abrogation of the simultaneous interactions mediated by BCL9/9L would influence the expression of genes associated with TBX3 peaks, we set out to mutate the *Bcl9/9l* interaction domains, while leaving TBX3 protein unaffected. We combined different *Bcl9/9l* alleles in which the HD2 (β-catenin-interacting) and HD1 (PYGO/new co-factor-interacting) motifs are

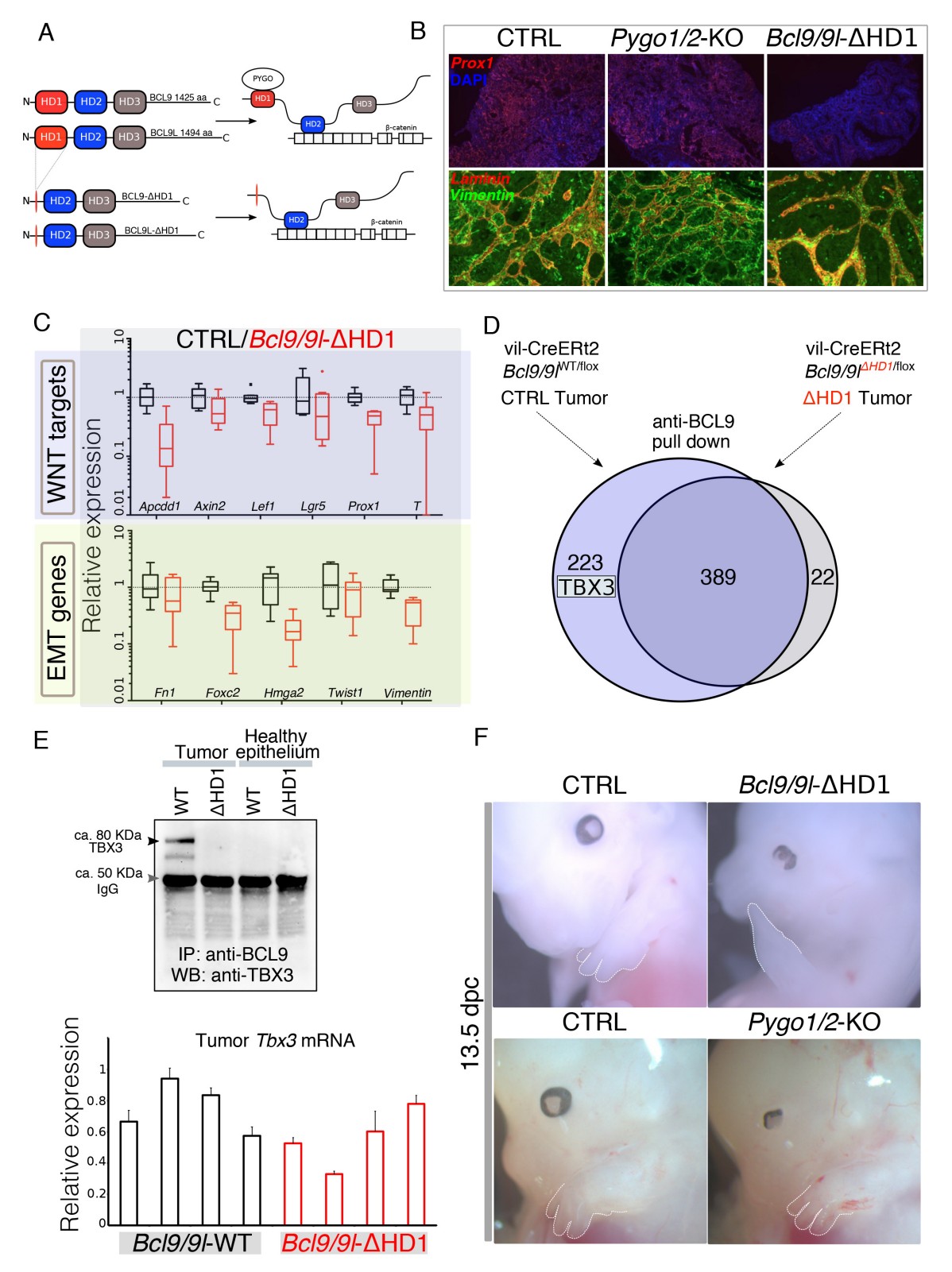

**Figure 2.** Identification of TBX3 as a putative BCL9 cofactor. (**A**) The deletion of the HD1 (PYGO-interacting) domain of BCL9 and BCL9L induces a variation in the 'chain of adaptors' causing the loss of PYGO association with the Wnt/β-catenin transcriptional complex (*Cantù et al., 2014*). (**B**) Immunofluorescence staining of tumors collected from control or conditional Pygo1/2-KO and *Bcl9/9l*-KO mice. Prox1 (red) and DAPI (blue) are shown in in the top panels; Vimentin (green) and Laminin (red) in the bottom panels. (**C**) Quantitative RT-PCR of selected groups of targets (compare it with *Figure 2 continued on next page*

*Figure 2 continued*

the same analysis of *Pygo1/2*-KO in *Figure 1E*) of RNA extracted from control or *Bcl9/9l*-ΔHD1 tumors. (**D**) Experimental outline of the tumor proteins pull-down and mass-spectrometry. TBX3 was identified among the proteins potentially interacting with BCL9 but not with BCL9-ΔHD1. (**E**) The IP proteins analyzed by mass spectrometry were in parallel subjected to SDS page electrophoresis and probed with an anti-TBX3 antibody (upper panel). The expression of *Tbx3* in control compared to *Bcl9/9l*-ΔHD1 tumors (N = 4) was evaluated via qRT-PCR (bottom panel), to exclude that differential pull-down was due to lost expression in mutant tumors. (**F**) 13.5 dpc *Bcl9/9l*-ΔHD1 embryos display forelimb malformations and absence of digits (emphasized by dashed white lines) – a characteristic *Tbx3*-mutant phenotype (upper panels). The limb defect is absent in *Pygo1/2*-KO embryos (bottom panels) underscoring that BCL9/9L act, in this context, independently of PYGO1/2.

The online version of this article includes the following figure supplement(s) for figure 2:

**Figure supplement 1.** qRT-PCR of target genes associated with intestinal stem cell function.
**Figure supplement 2.** Cytological proximity of BCL9 and TBX3.

---

deleted (*Cantù et al., 2018*). Double heterozygous animals for the HD1 (*Bcl9*$^{\Delta HD1/+}$; *Bcl9l*$^{\Delta HD1/+}$) or the HD2 (*Bcl9*$^{\Delta HD2/+}$; *Bcl9l*$^{\Delta HD2/+}$) deletions are viable and fertile. The cross between them leads to a trans-heterozygous genetic configuration in which both domain deletions are present (*Bcl9*$^{\Delta HD1/\Delta HD2}$; *Bcl9l*$^{\Delta HD1/\Delta HD2}$, referred to as *Bcl9/9l*-Δ1/Δ2, *Figure 1A*). As in these mice BCL9/9L retain both the HD1 and the HD2 domains in heterozygosity, this allelic combination is a way of testing the consequences of abrogating the tripartite complex mediated by the two interacting motifs of BCL9/9L without causing a full loss-of-function of these proteins. *Bcl9/9l*-Δ1/Δ2 embryos also display forelimb malformations, the cause of which cannot be due to PYGO (*Figure 2F*) but must be caused by the failure of recruiting the new HD1-interacting partner by BCL9/9L onto the β-catenin transcriptional complex. We collected forelimbs from control and *Bcl9/9l*-Δ1/Δ2 mutant embryos at 10.5, and measured gene expression via RNA-seq (*Figure 3G*). We found a significant enrichment (Hypergeometric test, p=1.4e-6) of TBX3 targets among the genes differentially expressed in *Bcl9/9l*-Δ1/Δ2 mutants (*Figure 3H*). The enrichment was particularly significant when considering down-regulated genes in *Bcl9/9l*-Δ1/Δ2 mutants, indicating that the BCL9-TBX3 partnership sustains transcriptional activation (*Figure 3H*). Of note, the design of our experiment directly implicates that these TBX3 transcriptional targets are also β-catenin-dependent. The overlap list includes several regulators of limb development, such as *Meis2* (*Capdevila et al., 1999*), *Irx3* (*Li et al., 2014*) and *Eya2* (*Grifone et al., 2007*; *Figure 3H*, heat map on the right). Despite being of correlative nature, this analysis supports a model in which BCL9/9L and TBX3 cooperate to the activation of target genes.

So far, we have presented genetic evidence that BCL9 proteins require additional co-factors, and that TBX3 associates with the β-catenin/BCL9 bound regions on the genome possibly influencing the expression of target genes. However, the similarity of genomic binding profiles between TBX3 and BCL9 might be due to their binding in different cells, and the decreased expression of genes with nearby enhancers bound by BCL9 and TBX3 might imply a requirement for BCL9/9L, but not necessarily for TBX3. We reasoned that our hypothesis - in which BCL9 functionally tethers TBX3 to the β-catenin transcriptional complex - raises several testable predictions that will be addressed below.

First, our model implies that TBX3 could impact on Wnt target gene expression and its activity should be dependent on the main constituents of the Wnt/β-catenin transcriptional complex. Second, if TBX3 is tethered by BCL9 to its targets, mutations in *BCL9/9L* should influence the ability of TBX3 to physically associate with WREs. Finally, as for BCL9, TBX3 should be capable of enhancing the metastatic potential of colorectal cancer cells.

To test our first prediction, implying a potential role of TBX3 in the transcription of Wnt target genes, we overexpressed it in HEK293T cells and monitored the activation status of Wnt signaling using the transcriptional reporter SuperTopFlash (STF). Consistent with its role as repressor, TBX3 led to a moderate but significant transcriptional downregulation that was, importantly, specific to the STF but not the control reporter plasmid (*Figure 4A*). Upon Wnt signaling activation achieved via GSK3 inhibition, TBX3-overexpressing cells exhibited a markedly increased reporter activity when compared to control cells, in particular at non-saturating pathway stimulating conditions (*Figure 4A*, left panel). Importantly, TBX3 proved transcriptionally incompetent on the STF if the cells carried mutations in *TCF/LEF* (Δ4TCF) or *CTNNB1* (encoding for β-catenin, Δβ-CAT; *Doumpas et al., 2019*), strongly supporting the notion of its cell-autonomous involvement in the activation of canonical Wnt target gene transcription (*Figure 4A*, central and right panels, respectively). Endogenous Wnt targets showed a similar expression behaviour to that of STF upon TBX3 overexpression (*Figure 4—*

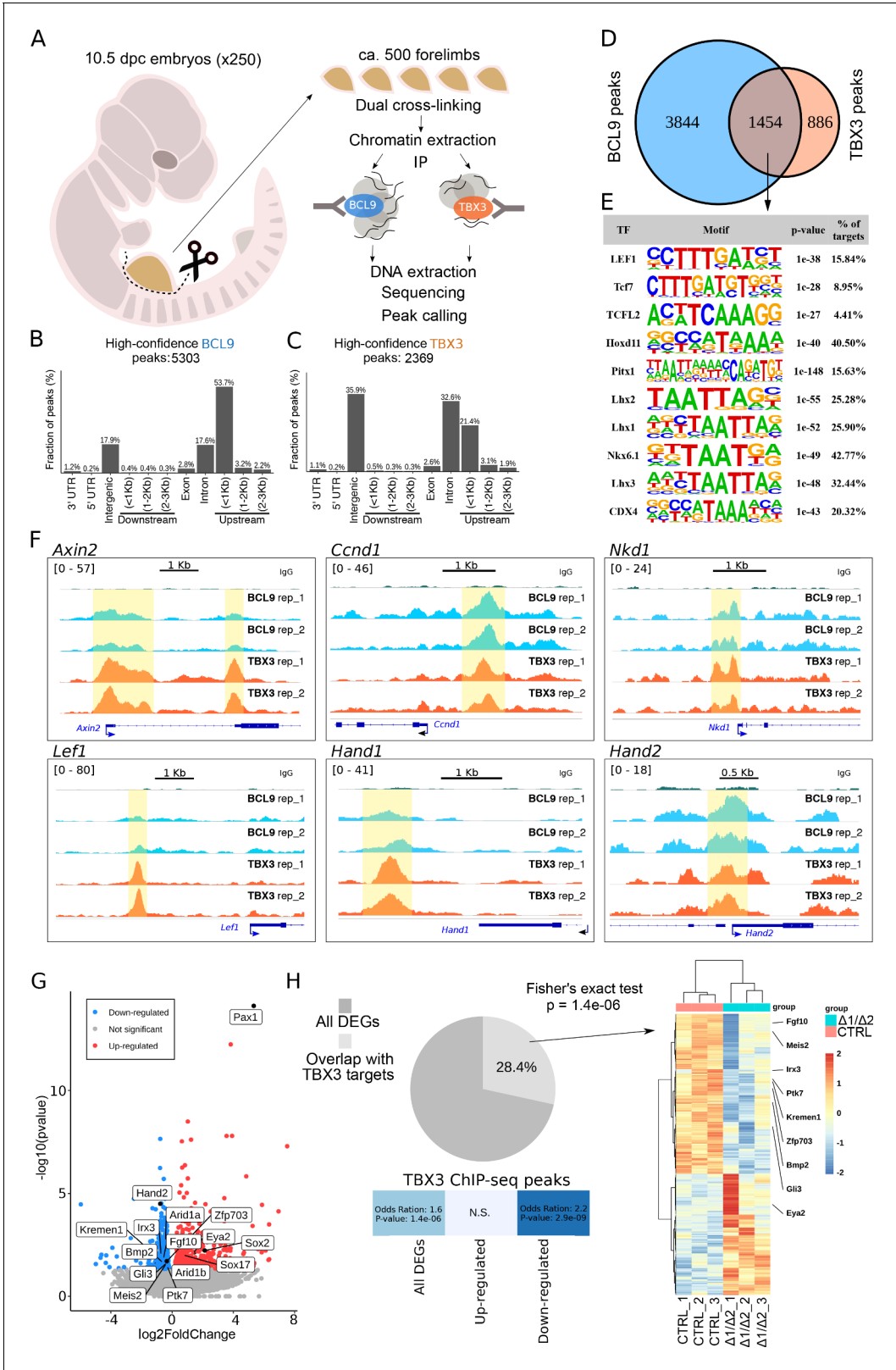

**Figure 3.** TBX3 and BCL9 occupy Wnt Responsive Elements (WRE) in vivo. (**A**) Artistic representation of the ChIP-seq experimental outline. (**B–C**) Bar-plots showing the genomic distribution of high-confidence BCL9 peaks (B, 5303 total) and TBX3 peaks (C, 2369 total). (**D**) Overlap of the high-confidence peak groups between BCL9 and TBX3. (**E**) Selected result entries from motif analysis performed on the BCL9-TBX3 overlapping high-confidence peaks. Significant enrichment was found for TCF/LEF and Homeobox motifs. No TBX consensus sequence was detected in this analysis. (**F**)

*Figure 3 continued on next page*

*Figure 3 continued*

Select genomic tracks demonstrating occupancy of BCL9 and TBX3 within the Wnt Responsive Element (WRE) of known Wnt-target genes (*Axin2*, *Ccnd1*, *Nkd1* and *Lef1*) and genes important in limb morphogenesis (*Hand1* and *Hand2*). The scale of peak enrichment is indicated in the top-left corner of each group of tracks. In light blue the BCL9 (*Salazar et al., 2019*) and in orange the TBX3 replicates, and in green the control track (IgG). Genomic tracks are adapted for this figure upon visualization with IGV Integrative Genomic Viewer (https://igv.org/). Two independent replicates for BCL9 and TBX3 ChIP-seq experiments are shown. (G) Volcano plot displays all the differentially expressed genes (DEGs) in developing forelimbs, upon mutation of *Bcl9/9l* (*Bcl9/9l-Δ1/Δ2* vs CTRL). DEGs were a total of 1143 (p<0.05), with 606 up-regulated and 537 down-regulated. N = 3 of individual mouse embryos for each condition were used for this analysis. (H) A significant portion (28.4%) of DEGs exhibited overlap with TBX3 ChIP-seq peaks. The overlap with TBX3 ChIP-seq peaks appeared statistically significant, in particular, when the down-regulated genes were considered. Hierarchical clustering of samples (3 CTRL versus 3 *Bcl9/9l-Δ1/Δ2*, right panel) based on genes overlapping between DEGs and genes annotated for TBX3 ChIP-seq peaks (normalized RNA-seq read counts, Ward's clustering method, Euclidian distance). Annotation added for genes associated by Gene Ontology to Wnt signaling (*Fgf10*, *Ptk7*, *Kremen1*, *Zfp703*, *Bmp2* and *Gli3*) and genes known as regulators of limb development (*Meis2*, *Irx3* and *Eya2*).

The online version of this article includes the following figure supplement(s) for figure 3:

**Figure supplement 1.** Overlap of the high-confidence BCL9 and TBX3 peaks in developing murine limbs reveals the existence of BCL9 exclusive (one example displayed in the genomic tracks on the left) and TBX3 exclusive peaks (one example in the genomic track on the right).

*figure supplement 1*). While our experiments show that TBX3 can influence the expression of Wnt target genes, the mechanisms by which this occurs remain to be elucidated.

We then addressed our second hypothesis, in which BCL9/9L are required for tethering TBX3 onto WREs. We performed ChIP of TBX3 in HEK293T cells followed by quantitative PCR to detect enrichment on the WRE present in the *AXIN2* promoter (*Jho et al., 2002*). Consistent with an effect on transcription in the absence as well as in the presence of Chir99021 (Chir) (*Figure 4A*), TBX3 was bound to this region both in 'OFF' and in 'ON' conditions (*Figure 4B*). We then exploited a HEK293T clone devoid of both BCL9 and BCL9L (ΔB9/9L; *van Tienen et al., 2017*), and tested if TBX3 was capable of physical association with the WRE. Of note, enrichment of TBX3 in ΔB9/9 L cells was dramatically reduced to background levels (*Figure 4B*). While we cannot exclude that TBX3 might act independently of BCL9/9L on several of its targets, this observation supports the notion that BCL9/9L are responsible for TBX3 recruitment on classical WREs (*Figure 4C*). This also suggests that the previously identified targets of both BCL9 and TBX3 (*Figure 3D–F*) must display simultaneous co-occupancy of these two factors. In agreement with the notion that BCL9/9L are themselves recruited by the TCF-β-catenin axis, the physical association of TBX3 with the *AXIN2* promoter was also lost in Δ4TCF and Δβ-CAT cells (*Figure 4B*).

Finally, we evaluated the effects of TBX3 overexpression (OE) on growth and metastatic potential of HCT116 human colorectal tumor cells – a representative model of CRC driven by activating mutations in *CTNNB1* (*Mouradov et al., 2014*) –, using a in vivo zebrafish xenograft model (*Rouhi et al., 2010*). Approximately 200–500 labelled control or TBX3-OE HCT116 cells were implanted in the perivitelline space of 72 hours post-fertilization (hpf) zebrafish embryos (*Figure 4D*). Three days after injection, TBX3-OE cells displayed a marked increase in number in the caudal hematopoietic plexus (*Figure 4E–F*), the main metastatic site for cells migrating from the perivitelline space (*Rouhi et al., 2010*). Of note, TBX3-OE HCT116 cells maintained consistently high expression of TBX3 within fish embryos throughout the experiment, and this was accompanied by increased Wnt/β-catenin-dependent transcription, as measured by *AXIN2* expression (*Figure 4G*). While this experiment does not allow to exclude that TBX3 might also act independently of BCL9/β-catenin in this context, it shows that increased expression of TBX3 enhances proliferation and migratory capability of human CRC cells bearing constitutively active Wnt signaling, and this is associated with simultaneous enhancement of the Wnt/β-catenin-dependent transcription (*Figure 4G*).

Taken together, our experiments show that, in specific developmental and disease contexts, the transcription factor TBX3 can take active part in the direct regulation of Wnt target genes by functional interplay with the β-catenin/BCL9-dependent transcriptional complex. Our study suggests a new paradigm in which tissue-specific co-factors might be the key to understand the spectrum of possible transcriptional outputs observed downstream of Wnt/β-catenin signaling (*Nakamura et al., 2016*). Moreover, TBX3 has been linked to different cancer types (*Willmer et al., 2017*). Our observations suggest that TBX3, or its downstream effectors, could be considered as new relevant targets to dampen CRC progression.

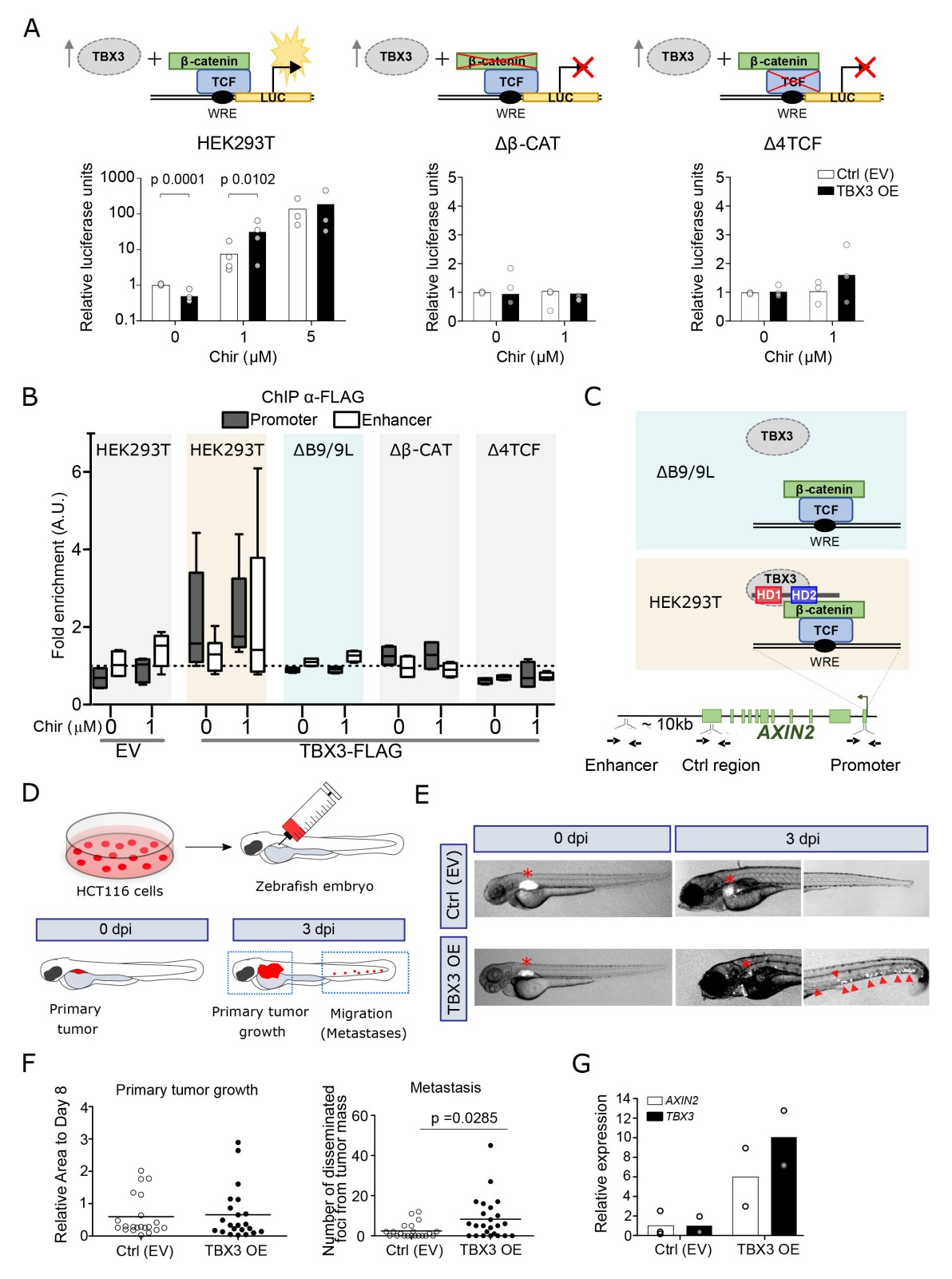

**Figure 4.** TBX3 controls the expression of Wnt target genes. (**A**) β-Catenin/TCF luciferase reporter STF assay in parental (left), β-catenin knockout (Δβ-CAT, center) and TCF knockout (Δ4TCF, right) HEK293T cells. Cells were treated with the indicated concentration of Chir or DMSO, overnight. Overexpression of TBX3 (OE, black bars) compared to control (EV, empty vector, white bars) showed that TBX3 acts as a repressor on a Wnt/TCF pathway reporter, but switches to activator upon pathway induction. Only significant p-values (p<0.05) are indicated. Three independent experiments

*Figure 4 continued on next page*

*Figure 4 continued*

(N = 3) are shown. Note that the logarithmic scale on the y-axis of the histogram on the left is different from the linear scale of the central and middle panels. (B) ChIP followed by qPCR in HEK293T cells treated with DMSO ('WNT-OFF') or Chir ('WNT-ON'). Enrichment was identified on *AXIN2* promoter and the downstream enhancer. Note that the enrichment on the enhancer is only present upon pathway stimulation: we interpret this as evidence for the enhancer looping onto the promoter occurring when the Wnt-dependent transcriptional regulation is active. The data are normalized to immunoprecipitation performed in cells transfected with an empty vector (EV) and presented as the mean ± standard deviation of independent experiments. The fold enrichment of TBX3-FLAG on *AXIN2* promoter and enhancer (N = 4) is lost upon mutations in *BCL9/9L* (ΔB9/9L, N = 2), *CNTTB1* (ΔB-CAT, N = 2) and *TCF/LEF* (Δ4TCF, N = 2). (C) Schematic representation of the *AXIN2* locus indicates the position of the primers used (black arrows) to test the binding of TBX3. Despite the apparent absence of direct physical interaction between TBX3 and BCL9/9L, the data support a model of TBX3 recruitment by BCL9/9L onto the β-catenin/TCF transcriptional complex. (D) Schematic diagram of the human CRC zebrafish xenografts model. Parental and TBX3- overexpressing HCT116 colorectal tumor cells were harvested and labeled with DiI dye (red). The stained cells were injected into the perivitelline space of 3 day old zebrafish embryos. Zebrafish were visualized with fluorescent microscopy at 0 day post injection (dpi) and three dpi, and primary tumor cell invasion and metastasis were counted. (E) Representative images of HCT116 tumor invasion and dissemination at 0 and 3 dpi in zebrafish xenografts, for both control and TBX3 overexpressing cells. The red asterisks indicate the position of the primary tumor. Red arrowheads point at clusters of disseminating/metastatic cells. (F) Scatter plot representing the quantification of primary tumor growth and metastasis after HCT116 xenograft. Horizontal bars represent the mean value. Only significant p-values (p<0.05) are displayed. (G) Quantitative RT-PCR confirmed continued increased expression of *TBX3* while HCT116 disseminate through zebrafish tissue, and that this is accompanied by enhanced Wnt/β-catenin transcription, as seen by *AXIN2* expression. Each datapoint represents the extraction of total RNA from pools of 10 zebrafish embryos. Figure Legend of Figure Supplements.

The online version of this article includes the following figure supplement(s) for figure 4:

**Figure supplement 1.** *AXIN2* and *NKD1* are here considered as representative Wnt transcriptional targets.

## Materials and methods

### Treatment of mice and histological analyses

Homeostasis: 4–6 week-old male and female mice (*Bcl9*$^{flox/flox}$;*Bcl9l*$^{flox/flox}$; vil-Cre-ERt2 and *Bcl9*$^{flox/flox}$;*Bcl9l*$^{flox/flox}$ (no Cre) littermates; *Pygo1*$^{flox/flox}$;*Pygo2*$^{flox/flox}$; vil-Cre-ERt2 and *Bcl9*$^{flox/flox}$;*Bcl9l*$^{flox/flox}$ (no Cre) were treated with five tamoxifen injections (i.p., 1 mg/day, Sigma) for five consecutive days and the small intestine and colon were analyzed at different time points thereafter. Mouse experiments were performed in accordance with Swiss guidelines and approved by the Veterinarian Office of Vaud, Switzerland.

Induction of DSS Colitis: 6–8 week-old male mice were treated with five tamoxifen injections (i.p., 1 mg/day) for five consecutive days. 10 days later 2.5% DSS (MG 36–50'000, MP Biomedicals, cat. no. 160110) was administered, ad libitum, in the drinking water for 9 days.

Induction of tumors: 6–8 week-old female mice were treated with five tamoxifen injections (i.p., 1 mg/day) for five consecutive days. 10 days later they were injected i.p. with 44 mg/kg body weight DMH 2HCl (N,N' Dimethylhydrazine dihydrochloride). After another 7 days later 2% DSS was administered, ad libitum, in the drinking water for 7 days.

Mice were monitored clinically for rectal bleeding, prolapse and general signs of morbidity, including hunched posture, apathetic behavior and failure to groom.

The relative body weight (in %) was calculated as follows: 100 X weight at a certain day/weight at the first day of DSS treatment. Epithelial damage of DSS treated mice was defined as percentage of distal colon devoid of epithelium.

To determine proliferation rates, mice were injected i.p. with 100 mg/kg BrdU (Sigma) 2 hr prior to sacrifice. Small intestines and colons (divided into three equal segments to be named proximal, middle and distal colon) were dissected, flushed with cold PBS, cut open longitudinally and fixed in 4% paraformaldehyde for 2 hr at RT and paraffin embedded. Sections (4 μm) were cut and used for hematoxylin/eosin and alcian blue staining and for immunohistochemistry. The primary antibodies used were rabbit anti-Synptophysin (DAKO; 1:100), rabbit anti-Lysozyme (1:500; DAKO), mouse anti-Ki67 (1:100; Novocastra), mouse anti-BrdU (1:500; Sigma), anti-β-Catenin (1:100; BD pharmigen), anti-active caspase 3 (1:100; Cell Signaling).

The peroxidase-conjugated secondary antibodies used were Mouse or Rabbit EnVision+ (Dako) or mouse anti-rat HRP (1:250: Biosource).

## Real-time PCR genotyping

To determine the deletion rate, the intestinal epithelium was separated from the underlining muscle. The intestine was dissected, flushed with PBS, cut open longitudinally and incubated in 3 mM ethylenediamine tetraacetic acid (EDTA) and 0.05 mM dithiothreitol (DTT) in PBS for 1.5 hr at RT on a rotor. The tubes were shaken vigorously, the muscle removed, and the epithelium centrifuged and used for genomic DNA extraction. SYBR green real-time PCR assays were performed on each sample analyzed.

## Chromatin immunoprecipitation

Forelimb buds were manually dissected from ca. 250 RjOrl:SWISS outbred 10.5 dpc mouse embryos. Chromatin immunoprecipitation was performed as previously described (*Cantù et al., 2018*). Briefly, the tissue was dissociated to a single cell suspension with collagenase (1μg/ml in PBS) for 1 hr at 37° C, washed and crosslinked in 20 ml PBS for 40 min with the addition of 1.5 mM ethylene glycol-bis (succinimidyl succinate) (Thermo Scientific, Waltham, MA, USA), for protein-protein crosslinking (*Salazar et al., 2019*), and 1% formaldehyde for the last 20 min of incubation, to preserve DNA-protein interactions. The reaction was blocked with glycine and the cells were subsequently lysed in 1 ml HEPES buffer (0.3% SDS, 1% Triton-X 100, 0.15 M NaCl, 1 mM EDTA, 0.5 mM EGTA, 20 mM HEPES). Chromatin was sheared using Covaris S2 (Covaris, Woburn, MA, USA) for 8 min with the following set up: duty cycle: max, intensity: max, cycles/burst: max, mode: Power Tracking. The sonicated chromatin was diluted to 0.15% SDS and incubated overnight at 4°C with 10 μg of anti-BCL9 (Abcam, ab37305) or anti-TBX3 (Santacruz, sc-17871) or IgG and 50 μl of protein A/G magnetic beads (Upstate). The beads were washed at 4°C with wash buffer 1 (0.1% SDS, 0.1% deoxycholate, 1% Triton X-100, 0.15 M NaCl, 1 mM EDTA, 0.5 mM EGTA, 20 mM HEPES), wash buffer 2 (0.1% SDS, 0.1% sodium deoxycholate, 1% Triton X-100, 0.5 M NaCl, 1 mM EDTA, 0.5 mM EGTA, 20 mM HEPES), wash buffer 3 (0.25 M LiCl, 0.5% sodium deoxycholate, 0.5% NP-40, 1 mM EDTA, 0.5 mM EGTA, 20 mM HEPES), and finally twice with Tris EDTA buffer. The chromatin was eluted with 1% SDS, 0.1 M NaHCO3, de-crosslinked by incubation at 65°C for 5 hr with 200 mM NaCl, extracted with phenol-chloroform, and ethanol precipitated. The immunoprecipitated DNA was used as input material for DNA deep sequencing. The pull downs of BCL9 and TBX3 were performed in parallel experiments. Note that the ChIP-seq of BCL9 has been already described in *Salazar et al., 2019*, and entirely re-analyzed here (see below).

Data analysis: Overall sequencing quality of the acquired fastq files was assessed using FastQC (version 0.11.5). Because all the samples exhibited good quality (MAPQ >30) and had no adapter contamination >0.1% trimming of reads was not deemed necessary. In addition, test alignments against several reference genomes were done using the FastQ Screen tool (version 0.13.0). Quality results were summarized using MultiQC (version 1.7). Fastq files were mapped to mouse reference genome (mm10) using the read aligner Bowtie2 (version 2.3.4.1). The mouse reference genome was downloaded from UCSC(http://hgdownload.cse.ucsc.edu/goldenpath/mm10/bigZips/). The resulting alignment files were then adjusted (conversion to binary format, removal of read aligned to mitochondrial DNA and indexing) using SamTools (version 1.9). To identify genomic regions enriched with aligned reads the peak calling tool MACS2 (version 2.2.6) was used. Calculated p-values were adjusted for false discovery rate (FDR) using Benjamini-Hochberg procedure, generating q-values. A cutoff q < 0.05 was used to assess significance. IgG sample was used as enrichment-normalization control. MACS2 generated peak files were further filtered by removal of blacklisted regions according to the ENCODE project using bedtools (version 2.26.0). Annotation and visualization were made with the R programming language (version 3.4.4) and Rstudio (version 1.1.463), using R packages ChIPpeakAnno (version 3.12.7), ChIPseeker (version 1.14.2) and ggplot2 (version 3.2.1). Genomic track visualization was done with Integrative Genomics Viewer (IGV) (version 2.4.17). Motif analysis was performed with HOMER. The data have been deposited at ArrayExpress with accession number E-MTAB-8997.

## RNA-seq data analysis

Quality control of fastq files was done using FastQC (version 0.11.5). Trimming of reads to remove adapter remnants and low quality read (MAPQ <30) was performed with BBDuk, part of the BBTools

suite (version 38.58). Test alignments against different reference genomes were done with the FastQ Screen tool (version 0.13.0).

Quality results were summarized using MultiQC (version 1.7). Sequenced reads were mapped to mouse reference genome (mm10) using the Spliced Transcript Alignment to a Reference (STAR) software (version 2.7.3a). Reference genome data (FASTA and GTF) were downloaded from GENCODE, release M24 (https://www.gencodegenes.org/mouse/). Downstream analyses, annotation and visualization were made with the R programming language (version 3.4.4) and Rstudio (version 1.1.463). Differentially expressed genes were assessed using the DESeq2 package (version 1.18.1). Significance was determined based on false discovery rate (FDR) adjusted p<0.05. Hierarchical clustered heatmap was produced with the pheatmap R package, using Ward's Hierarchical Agglomerative Clustering Method. A complete list of bioinformatic tools and references is listed in the accompanying *Supplementary file 1*. The RNA-seq experiment has been deposited at ArrayExpress with accession number E-MTAB-9000.

## Protein immunoprecipitation and mass spectrometry

Dissected mouse tumors were minced in cold PBS and treated with a hypotonic lysis buffer (20 mM tris-HCl, 75 mM NaCl, 1.5 mM MgCl2, 1 mM EGTA, 0.5% NP-40, and 5% glycerol). Protein extracts obtained were incubated with 1 µg of anti-BCL9 antibody (ab54833 or ab37305, Abcam) and protein A–conjugated Sepharose beads (GE Healthcare); they were then diluted in lysis buffer to a final volume of 1 ml. After 4 hr of incubation at 4°C on a rotating wheel, the beads were spun down and washed three times in lysis buffer. All steps were performed on ice, and all buffers were supplemented with fresh protease inhibitors (cOmplete, Roche) and 1 mM phenylmethylsulfonyl fluoride. For detecting the proteins in Western blot, the pulldown reactions were treated with Laemmli buffer, boiled at 95°C for 15 min, and subjected to SDS-PAGE separation and blotting on a polyvinylidene difluoride (PVDF) membrane. The PVDF membrane was probed with the anti–TBX3.

For liquid chromatography–MS/MS analysis, the protein samples, already dissolved in Laemmli buffer, were submitted to a filter-aided sample preparation (FASP) and digested with trypsin in 100 mM triethylammonium bicarbonate buffer overnight. Desalted samples were dried completely in a vacuum centrifuge and reconstituted with 50 µl of 3% acetonitrile and 0.1% formic acid. Each peptide solution (4 µl) was analyzed on both Q Exactive and Fusion mass spectrometers (Thermo Scientific) coupled to EASY-nLC 1000 (Thermo Scientific). Spectra acquisition and peptide count were performed as described in details in *Cantù et al., 2017*. The dataset is deposited at the PRIDE repository under accession number PXD018805.

## Cell culture

The human cell lines HEK293T -parental, ΔCTNNB1 and ΔTCF/LEF (*Doumpas et al., 2019*), BCL9/9L double KO HEK293T (*van Tienen et al., 2017*), and HCT116 were cultured in DMEM (Thermo Fisher Scientific, Belmont, Massachusetts, US) supplemented with 10% fetal bovine serum (Gibco, Gaithersburg, USA) 1% L-glutamine and 1% penicillin-streptomycin at 37°C in a humidified chamber supplemented with 5% CO2. Chir99021 compound (Tocris Bioscience) was used as Wnt signaling activator. At 6 hr after transfection, the transfected cells were treated for 16 hr either with DMSO (0 µM Chir), 1 µM or 5 µM Chir. Following that, cells were collected for further experiments. All cell lines were tested and scored negative for mycoplasma infection. HCT116 were obtained from DSMZ (https://www.dsmz.de/collection/catalogue/details/culture/ACC-581) and kindly donated by Prof. Stefan Koch.

## Plasmids and transfection

TBX3-3xflag (donated by Peter J. Hurlin), pCMV-Pygo2-HA (*Cantù et al., 2013*), pCDNA3.1-BCL9-GFP (kindly gifted by Mariann Bienz), pcDNA3-eGFP (kindly provided by Stefan Koch), M50 SUPER 8X TOPFLASH pTA-Luc (Addgene 12456) and M51 SUPER 8X FOPFLASH pTA-Luc (Addgene 12456) were transfected using Lipofectamine 2000 (Invitrogen, USA) following the manufacturer's instructions. Twenty-four hours after transfection, the cells were collected for the indicated experiments.

## RNA extraction and quantitative reverse transcriptase PCR (qRT-PCR)

Total RNAs from cells were extracted using TRIzol (Thermo Fisher Scientific, Belmont, Massachusetts, US) following the manufacturer's instructions. After reverse transcription reaction with High Capacity cDNA Reverse Transcription Kit (Thermo Fisher Scientific), qPCR was conducted with SYBR green LightCycler 480 (Roche) on CFX96 Real-Time PCR Detection System (Bio-Rad, USA). *GAPDH* was used as endogenous control and the relative expression of RNAs was calculated using the $2^{-\Delta\Delta Ct}$ method. The primers used in this study were designed by Primer3plus web and their sequences are listed in the *Supplementary file 2*.

## Xenograft tumor zebrafish model

HCT116 cells were transfected with pCS2 (empty vector) or TBX3-expressing plasmid and labeled with 1,1'-dioctadecyl-3,3,3',3'-tetramethylindocarbocyanine perchlorate (DiI, cat. no. D3899; Invitrogen) according to previously described method (*Rouhi et al., 2010*). Briefly, tumor cells were washed twice with PBS, followed by labeling with a final concentration of 5 µg/ml of DiI for 30 min at 37 ˚C. After rigorous washing with PBS, tumor cells were trypsinized for 5 min, counted under a phase contrast microscope, centrifuged at 300 g for 5 min and resuspended at a final concentration of 30–40 cells per µl in medium. Human cells were injected into the perivitelline space of 72 hr old fli1:EGFP transgenic zebrafish embryos. The eggs were fertilized, collected and dechorionated. After that, HCT116 cells were injected into the vitreous cavity of zebrafish embryo with non-filamentous borosilicate glass capillary needles attached to the microinjector under the stereomicroscope (Leica Microsystems). After tumor cell injection, zebrafish embryos were further selected under fluorescent microscopy to ensure that tumor cells were located only within the cavity and then incubated in aquarium water for consecutive 3 days at 36.0 ˚C. Primary tumor growth, invasion and metastasis in the zebrafish body were monitored at day three with a fluorescent microscope (Nikon Eclipse C1) as previously published (*Rouhi et al., 2010*). Briefly, each zebrafish embryo was picked up and monitored to detect tumor cell distribution. Two different sets of images from the head region and the trunk region were collected separately from each zebrafish embryo. Disseminated tumor cells in the caudal hematopoietic plexus of zebrafish embryos were counted (in a double-blind manner) and the primary tumor areas were measured using Image J software. At least 15–20 embryos were included in each experimental group.

## Western blot analysis

Cells were lysed at 4˚C for 10 min using NP-40 lysis buffer (50 mM Tris-HCl (pH 7.5), 120 mM NaCl, and 1% (v/v) Igepal), supplemented with protease inhibitor cocktail (Merck). Cells were sonicated with Q700CA Sonicator (Q Sonica) on 50 amplitude for 5 s on/off for two cycles. Protein lysates were separated by 10% SDS-polyacrylamide gel electrophoresis (SDS-PAGE) and transferred onto nitrocellulose membrane. The membrane was blocked in Odyssey TBS Blocking Buffers (LI-COR Bioscience) for 1 hr at room temperature and subsequently incubated with specific antibody against FLAG (1:1000, Merck F3165) overnight at 4˚C and HA (1:1000, Merck 05–902R) and GFP (1:1000 Santa Cruz Biotechnology sc-9996) 1 hr at room temperature. Afterwards, the membrane was washed and incubated with fluorescent Goat-anti-Rabbit and goat-anti-Mouse (1:2000; LI-COR Bioscience) for 1 hr at room temperature. Protein detection was performed in LI-COR Odyssey CLx (LI-COR Bioscience).

## Super-Top/Fop-Flash reporter assay

The β-catenin reporter plasmid (SuperTopFlash, STF) and its mutant control (Fop-flash) were constructed by Addgene (12456). Cells were plated in triplicate in a 96 well plated overnight and co-transfected with 50 ng Top flash or Fop flash expression plasmids together with other plasmids for the experiments using Lipofectamine 2000. The activities of firefly luciferase reporters were determined at 24 hr after transfection using ONE-Step Luciferase Assay System Kit (Pierce) according to the manufacturer's instructions. The Top/Fop ratio was used as a measure of β-catenin-driven transcription. GraphPad Prism software was used for statistical analyses. Values were expressed as mean ± standard deviation. Mann Whitney U test was used to analyze the differences between two groups and $p < 0.05$ was considered statistically significant. All experiments were performed at least three times and all samples analyzed in triplicate unless otherwise stated.

## In-situ proximity ligation

Proximity ligation was performed using NaveniFlex MR assay reagents according to the supplier's guidelines (Navinci Diagnostics AB). In brief, HEK293T cells were plated on coverslips and were transiently transfected using lipofectamine 2000 transfection reagent (ThermoFisher Scientific) with plasmids expressing the following combinations of proteins (each in a ration 1:1): BCL9-HA and TBX3-FLAG, BCL9-HA and GFP or BCL9-GFP and PYGO2-HA. An EGFP-expressing vector at 1/10th the total plasmid amount was also added in PLA assays to identify transfected cells. Paired combinations of rabbit (anti-HA) and mouse (anti-FLAG or anti-GFP) antibodies were incubated with samples for 1 hr at room temperature (1:1000 dilution each). Following addition of matched PLA probes, rolling circle amplification proceeded in presence of TEX 615 fluorophore. After washing, coverslips were mounted in Prolong Glass Antifade with NucBlue counterstain. Slides were imaged in Zeiss LSM770 confocal microscope, using 20x and 40x objective and filters set in DAPI, FITC and Texared. Specific individual protein-protein interactions can be seen as red dots. For all experimental conditions, at least three images were acquired.

Image analysis and quantification were performed using the opensource ImageJ2 software. The pipeline for signal quantification was the following: images in the green and red channels were merged and converted to a binary image (8-bit). A threshold range was set to outline the objects of interest (particles in yellow, GFP + PLA signal) and separate it from background signal. Positive dots representing the PLA signal 'particles' were counted using the command 'analyze particles'. The image in the blue channel (DAPI) was used to count cell nuclei. After converting the image to binary by thresholding binary conversion, overlapping objects were separated using the plugin Watershed and counted using the command 'analyze particles'. The minimum size of 50 micron$^2$ was set to conservatively exclude unspecific signals.

## Acknowledgements

The authors thank Peter J Hurlin for kindly donating the *TBX3* expressing plasmid, Mariann Bienz for donating the BCL9-GFP plasmid, George Hausmann, Stefan Koch, Lavanya Moparthi for sharing reagents and critical input, and Lasse Jensen for helping with zebrafish injection. We are also grateful to Mariann Bienz and Melissa Gammons for donating – and shipping very rapidly – the BCL9/9L-KO HEK293T cell clone. CC is a Wallenberg Molecular Medicine fellow and receives generous financial support from the Knut and Alice Wallenberg Foundation. This work was supported by Cancerfonden to CC (CAN 2018/542), by the Swiss National Science Foundation and the Canton of Zurich to KB and by the Swiss National Science Foundation (grant PCEPP3_181249) to AEM.

## Additional information

### Funding

| Funder | Grant reference number | Author |
| --- | --- | --- |
| Knut och Alice Wallenbergs Stiftelse | | Claudio Cantù |
| Cancerfonden | CAN 2018/542 | Claudio Cantù |
| Schweizerischer Nationalfonds zur Förderung der Wissenschaftlichen Forschung | PCEPP3_181249 | Andreas E Moor |
| Schweizerischer Nationalfonds zur Förderung der Wissenschaftlichen Forschung | | Konrad Basler |

The funders had no role in study design, data collection and interpretation, or the decision to submit the work for publication.

### Author contributions

Dario Zimmerli, Conceptualization, Data curation, Formal analysis, Validation, Investigation, Methodology, Writing - review and editing; Costanza Borrelli, Amaia Jauregi-Miguel, Data curation, Formal

analysis, Validation, Investigation, Methodology, Writing - review and editing; Simon Söderholm, Resources, Data curation, Software, Formal analysis, Investigation, Writing - review and editing; Salome Brütsch, Nikolaos Doumpas, Jan Reichmuth, Investigation, Methodology, Writing - review and editing; Fabienne Murphy-Seiler, Data curation, Formal analysis, Investigation, Methodology, Writing - review and editing; MIchel Aguet, Data curation, Supervision, Investigation, Methodology, Writing - review and editing; Konrad Basler, Conceptualization, Resources, Data curation, Supervision, Funding acquisition, Validation, Visualization, Project administration, Writing - review and editing; Andreas E Moor, Claudio Cantù, Conceptualization, Resources, Data curation, Formal analysis, Supervision, Funding acquisition, Validation, Investigation, Visualization, Methodology, Writing - original draft, Project administration, Writing - review and editing

### Author ORCIDs
Amaia Jauregi-Miguel (ID) https://orcid.org/0000-0003-0938-7734
Simon Söderholm (ID) http://orcid.org/0000-0001-5350-7102
Andreas E Moor (ID) https://orcid.org/0000-0001-8715-8449
Claudio Cantù (ID) https://orcid.org/0000-0003-1547-5415

### Ethics
Animal experimentation: Tumor and colitis induction experiments in the mouse were performed in Lausanne, in accordance with Swiss guidelines and approved by the Veterinarian Office of Vaud, Switzerland. Embryological studies were performed in Linköping, Sweden, under the ethical animal work license obtained by C.C. at Jordbruksverket (Dnr 2456-2019).

### Decision letter and Author response
Decision letter https://doi.org/10.7554/eLife.58123.sa1
Author response https://doi.org/10.7554/eLife.58123.sa2

# Additional files

### Supplementary files
• Supplementary file 1. List of the key computational resources used for RNA-seq and ChIP-seq analyses.
• Supplementary file 2. Sequences of primers used for qPCR and qRT-PCR reactions.
• Transparent reporting form

### Data availability
The ChIP-seq data have been deposited at ArrayExpress with accession number E-MTAB-8997. The RNA-seq experiment has been deposited at ArrayExpress with accession number E-MTAB-9000.

The following datasets were generated:

| Author(s) | Year | Dataset title | Dataset URL | Database and Identifier |
| --- | --- | --- | --- | --- |
| Cantù C, Zimmerli D | 2020 | RNA-seq experiment of developing mouse forelimbs, comparing Bcl9/9l mutant with control littermates. | https://www.ebi.ac.uk/arrayexpress/experiments/E-MTAB-9000/ | ArrayExpress, E-MTAB-9000 |
| Cantù C, Zimmerli D | 2020 | ChIP-seq experiment of the beta-catenin co-factor BCL9 and the T-box transcription factor TBX3 in developing mouse forelimbs. | https://www.ebi.ac.uk/arrayexpress/experiments/E-MTAB-8997/ | ArrayExpress, E-MTAB-8997 |
| Cantù C, Moor A | 2020 | Pull down of full length and mutant BCL9 (deltaHD1) in mouse colorectal tumors and colonic epithelium | https://www.ebi.ac.uk/pride/archive/projects/PXD018805 | PRIDE, PXD018805 |

The following previously published dataset was used:

| Author(s) | Year | Dataset title | Dataset URL | Database and Identifier |
|---|---|---|---|---|
| Cantù C, Zimmerli D | 2019 | ChIP-seq experiment of the beta-catenin co-factor Bcl9 in developing forelimbs. | https://www.ebi.ac.uk/arrayexpress/experiments/E-MTAB-7652 | ArrayExpress, E-MTAB-7652 |

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
