## [Decision Letter]

**Acceptance summary:**

This work analyzes the transcriptional output of the Wnt signaling pathway. Through a series a genetic and biochemical experiments, these results suggest that the TBX3 protein acts as a context-dependent component of the transcriptional complex. This is interesting as TBX3 was previously implicated as a target of the Wnt pathway. By also playing a role in the transcriptional complex activated by Wnt, TBX3 may be involved in a transcriptional circuit.

**Decision letter after peer review:**

Thank you for submitting your article "TBX3 acts as tissue-specific component of the Wnt/β-catenin enhanceosome" for consideration by *eLife*. Your article has been reviewed by three peer reviewers, one of whom is a member of our Board of Reviewing Editors, and the evaluation has been overseen by Kevin Struhl as the Senior Editor. The reviewers have opted to remain anonymous.

The reviewers have discussed the reviews with one another and the Reviewing Editor has drafted this decision to help you prepare a revised submission.

Your manuscript builds upon the observation that *Bcl9*/*Bcl9L* mutants have a more severe intestinal and limb defect than *Pygo1/2* double mutants. This is interesting because the *Bcl9s* and Pygos are often thought to act in a complex to regulate Wnt targets. A possible binding partner of *Bcl9* is TBX3, a gene implicated in limb deformities that might be similar to *Bcl9/9L* limb conditional knockouts. Using ChIP-seq of embryonic mouse limbs, your show co-localization of TBX3 and *Bcl9* at over 1400 locations (2/3s of the total TBX3 peaks). This leads to a model where TBX3 is recruited to Wnt target gene enhancers through interaction with *Bcl9/9L*.

The reviewers had a number of suggestions that would improve the manuscript. Some of these require additional experiments but the reviewers thought that these experiments would not require too much effort and time, even under the present COVID circumstances.

There is no direct evidence for co-occupancy, and the term must be changed. The binding profiles of *Bcl9* and TBX3 are similar, but co-occupancy means binding of both at the same time, not just the same place. The convincing experiment here would be sequential ChIP (re-ChIP). Moreover, the authors haven't demonstrated a physical interaction between the 2 proteins, which would at least add evidence (though not proof) for co-occupancy.

Specifically, the data in Figure 4A,B do not address the model favored by you, i.e., that *Bcl9/9L* recruits TBX3 to enhancers. Rather it shows that genes with nearby enhancers bound by *Bcl9* and TBX3 have decreased expression in *Bcl9/9L* mutants. This speaks to a requirement for *Bcl9/9L*, but doesn't address its relationship to TBX3. This can be tested by whether loss of *Bcl9/9L* affects TBX3 binding to enhancers. You will have the reagents to do this key experiment. In addition, it is suggested to test whether the ChIP signal at the Axin2 enhancer and promoter are dependent on β-catenin and TCFs, using the CRISPR lines used in Figure 4C. These experiments directly test their model with reagents that you have in hand. The reviewers also suggest that a proximality labeling experiment which can capture weaker interactions than co-IP could be done in Wnt-active versus inactive contexts to potentially confirm TBX3-*Bcl9* interaction and identify other cofactors.

The reviewers and editors point out that the authors should eliminate the term "enhanceosome" in the title and anywhere else. "Enhanceosome" has a very specific meaning in the transcription field, namely a highly structured physical entity composed of multiple transcription factors interacting with a complex and specific arrangement of target sites.

A reviewer's comment on phenotypes in seen *Pygo1/2*-KO and CTRL in regard to the length of villus and depth of crypts as well as number of proliferating cells. Therefore, in the absence of more detailed morphologic and quantitative histologic information, the description of the figure should be changed from "Intestinal architecture was normal" to "Not overt phenotypic defects".

Reviewer #1:

This manuscript builds upon the interesting observation that *Bcl9*/*Bcl9L* mutants have a more severe intestinal and limb defect than *Pygo1/2* double mutants. This is interesting because the *Bcl9s* and Pygos are often thought to act in a complex to regulate Wnt targets. Using *Bcl9* as bait, the authors then identify TBX3 as an interaction partner. TBX3 has limb deformities that might be similar to Bcl9/9L limb conditional knockouts. Using ChIP-seq of embryonic mouse limbs, they show an impressive co-localization of TBX3 and *Bcl9* at over 1400 locations (2/3s of the total TBX3 peaks). Finally, they attempt to support a model where TBX3 is recruited to Wnt target gene enhancers through interaction with *Bcl9/9L*.

The strengths of the manuscript are the convincing data demonstrating that *Bcl9/9L* mutants have more dramatic phenotypes than *Pygo1/2*. The ChIP-seq with *Bcl9* and TBX3 is also very interesting. But the data functionally linking TBX3 to *Bcl9* on Wnt targets is very weak and don't adequately support the molecular model that they present.

1) It's not clear how similar that limb defects in *Bcl9/9L* mutants are compared to the TBX3 mutant limbs described in Frank et al., 2013. I disagree that they are "strikingly similar". Without more direct experimental observation, this connection needs to be related with more careful language.

2) The data in Figure 4A,B does not address the model favored by the authors, i.e., that *Bcl9/9L* recruits TBX3 to enhancers. Rather it shows that genes with nearby enhancers bound by *Bcl9* and TBX3 have decreased expression in *Bcl9/9L* mutants. This speaks to a requirement for *Bcl9/9L*, but doesn't address its relationship to TBX3.

3) A direct test of their model would be to see whether loss of *Bcl9/9L* affects TBX3 binding to enhancers. The authors have the reagents to do this key experiment.

4) The overexpression data with TBX3 is unconvincing. At the concentration used, TBX3 represses Wnt targets in the absence of signaling. It's impossible to tell whether this is a direct or indirect affect (note the TBX3 ChIP data in Figure 4D indicates TBX3 is bound to the Axin2 promoter in the absence of signaling). The very small effect of TBX3 on TopFlash or Axin2 and Nkd1 is only "significant" when this repression is taken into account. This data is not acceptable evidence that TBX3 potentiates the expression of Wnt targets.

5) The zebrafish metastasis data is interesting, but it's unclear whether the ability to spread from sight of injection is dependent on Wnt signaling. So it's impossible to interpret whether it supports that authors' model.

6) They should test whether depletion of TBX3 affects Wnt regulation of Axin2 (their ChIP data in Figure 4D suggests Axin2 might be a direct TBX3 target). They should also test whether the ChIP signal at the Axin2 enhancer and promoter are dependent on β-catenin and TCFs (with the CRISPR lines they use in Figure 4C). These experiments directly test their model with reagents that they already have in hand.

Reviewer #2:

Aberrant activation of the canonical Wnt/β-catenin pathway drives the formation of various human cancers and has emerged as a promising, yet challenging, target for cancer therapy. BCL9/9L and PYGO1/2 are transcriptional co-activator's of β-catenin and it is believed that their concerted action is required for efficient activation of Wnt/β-catenin-target gene expression and tumor phenotype. In this manuscript, by using various strains of engineered mice, Zimmerli et al. provide genetic evidence indicating that BCL9/9L function does not entirely depend on PYGO1/2. They also showed that during forelimb development, BCL9/9L have a PYGO1/2 independent function, and identified TBX3 as a candidate tissue-specific transcriptional factor that functional interact with BCL9/9L in mediating Wnt/β-catenin transcriptional output.

This is a clear, well written manuscript, with experimentally supported conclusions, which can have important basic, translational and clinical implications.

1) The histologic characterization of CTRL and *Pygo1/2*-KO intestine and colon should be more systematic, detailed and quantified using digital pathology (e.g. Halo Analysis platform). From the representative pictures provided in Figure 1B and Figure 1—figure supplement 1C it seems that the villi and crypts are slightly shorter and the number of Ki-67 positive cells is reduced in *Pygo1/2*-KO mice.

2) In Figure 2E, because the epitope recognized by the BCL9 antibody could be masked during potential BCL9/TBX3 direct interaction, this reviewer is wondering whether the IP should be done with two anti-BCL9 antibodies, and a reverse IP using anti TBX3 antibodies should be also used.

3) In Figure 3F, it would be important to know what happens with the co-occupancy of BCL9 and TBX3 within the Wnt responsive element of Lgr5.

4) In legend of Figure 4, the label (D) before (C) should be eliminated or changed according to other labels.

Reviewer #3:

Nuclear translocation of β-catenin has been established as the necessary step in Wnt pathway activation and target gene transcription. However, relatively little is understood about how this component can induce specific, context-dependent transcriptional responses. Indeed, β-catenin interacts with many more genomic loci than the number of Wnt-regulated genes, indicating that β-catenin association with cis-regulatory elements might not be sufficient for Wnt target transcription (Nakamura et al., 2016) and that other transcriptional regulators might be involved. This study identifies TBX3 as one such novel cofactor for β-catenin mediated Wnt response.

Loss of Pygopus did not phenocopy deletion of *Bcl9* or the Pygopus-interacting HD1 domain of *Bcl9*, suggesting that additional factor(s) might be involved in *Bcl9* function. IP followed by mass spec in intestine tumors identified TBX3, whose interaction with *Bcl9* depends on the *Bcl9* HD1 domain. TBX3 and *Bcl9* ChIP-seq showed overlap among select Wnt responsive elements and TBX3 targets were enriched among genes downregulated upon deletion of the putative TBX3 binding domain of *Bcl9*. Two functional assays demonstrated role of TBX3 in Wnt target transcription and tumorigenesis in Wnt activated contexts: (1) TBX3 overexpression led to higher Wnt reporter induction in a manner dependent on pathway activation and β-catenin/Tcf and (2) TBX3 overexpression enhanced proliferation and migration of Wnt-dependent human colorectal tumor cells in a zebrafish xenograft assay.

This study identifies TBX3 , a transcription factor with known developmental roles, as a novel transcriptional co-regulator of Wnt targets in contexts such as colorectal cancer and limb development. It is a methodical and convincing investigation of TBX3 in multiple Wnt-activated contexts as is.

Some follow up questions include the extent of TBX3 involvement in activation of Wnt target genes in other contexts, and whether that depends on *Bcl9*. Does increased STF induction or tumorigenesis upon TBX3 overexpression require *Bcl9*, for instance? Does loss of TBX3 itself affect Wnt target activation? It would also be interesting to further probe TBX3-*Bcl9* or TBX3-β-catenin interaction. An RNA-seq experiment with *Bcl9* missing HD1 but intact HD2 domain, or TBX3 ChIP-seq in the context of *Bcl9* HD1 deletion, would further support functional interaction of *Bcl9* HD1 and TBX3 in Wnt target activation. Additionally, a proximality labeling experiment that can capture weaker interactions than co-IP could be done in Wnt-active versus inactive contexts to potentially confirm TBX3-*Bcl9* interaction and identify other cofactors.

---

## [Author Response]

Revisions for this paper:The reviewers had a number of suggestions that would improve the manuscript. Some of these require additional experiments but the reviewers thought that these experiments would not require too much effort and time, even under the present COVID circumstances.There is no direct evidence for co-occupancy, and the term must be changed. The binding profiles of Bcl9 and TBX3 are similar, but co-occupancy means binding of both at the same time, not just the same place. The convincing experiment here would be sequential ChIP (re-ChIP). Moreover, the authors haven't demonstrated a physical interaction between the 2 proteins, which would at least add evidence (though not proof) for co-occupancy.

We dedicated great efforts to address the concept of co-occupancy during the revision of our manuscript. We agree that ChIP-re-ChIP would have constituted convincing evidence. However, our re-ChIP attempts were inconclusive, and this was not entirely surprising due to the challenging nature of this experiment. However, we have obtained an equally informative observation by performing new ChIP experiments of TBX3 in a *BCL9/9L* mutant background. We exploited a HEK293T cell clone previously generated by van Tienen and colleagues (kindly provided by Mariann Bienz) in which *BCL9* and *BCL9L* have been deleted. Using this tool, we could now show that TBX3 fails to associate with the *AXIN2* promoter and enhancer in the absence of these proteins (Figure 4B of the revised manuscript). This is a very important new addition, as the most likely explanation of this observation is that BCL9 is required for TBX3 recruitment on Wnt Responsive Elements. Consistently, and in agreement with our proposition, the TBX3 ChIP signal is also lost in cell clones lacking β-catenin or the TCF/LEF transcription factors (an additional experiment we made to fulfill the reviewers’ requests).

Nevertheless, as we think this is a relevant aspect in the field of transcriptional regulation, we changed the term “co-occupy” into “occupy” throughout the description of our original ChIP-seq experiment, and made clear in the main text that this experiment does not prove co-occupancy.

Specifically, the data in Figure 4A,B do not address the model favored by you, i.e., that Bcl9/9L recruits TBX3 to enhancers. Rather it shows that genes with nearby enhancers bound by Bcl9 and TBX3 have decreased expression in Bcl9/9L mutants. This speaks to a requirement for Bcl9/9L, but doesn't address its relationship to TBX3. This can be tested by whether loss of Bcl9/9L affects TBX3 binding to enhancers. You will have the reagents to do this key experiment. In addition, it is suggested to test whether the ChIP signal at the Axin2 enhancer and promoter are dependent on β-catenin and TCFs, using the CRISPR lines used in Figure 4C. These experiments directly test their model with reagents that you have in hand.

We have now removed the emphasis previously put on the description of the RNA-seq experiment previously presented in Figure 4A-B (now in revised Figure 3G-H) and specified in the text that this experiment is of “correlative nature” and does not necessarily imply requirement of TBX3 for the expression of BCL9 targets.

On the other hand, we wish to point out that this is not a “simple” loss-of-function of *Bcl9/9l*. As now better explained in the main text, in the *Bcl9/9l*-Δ1/Δ2 embryos used for this analysis, each allele encoding for BCL9/9L retains either the HD1 or the HD2 domain. Some BCL9/9L proteins can use the HD1 for interacting (but not the HD2) and others still possess the HD2 (but not the HD1); however, no single BCL9/9L molecule could interact via the two domains simultaneously. Therefore, this allelic combination is a specific way of testing the consequences of abrogating the tripartite complex mediated by the two interacting domains of BCL9/9L. As we show that loss of *Pygo1/2* does not cause malformed forelimbs, the gene expression changes observed must be caused by whatever other factor fails to be tethered by BCL9/9L onto the β-catenin transcriptional complex. We agree however, that this does not imply that this co-factor is TBX3. The limitation of this experiment, rightfully emphasized by your comments, also constituted part of the impetus that motivated us in performing the new set of ChIP experiments explained before, which now clearly show that the physical association of TBX3 with the AXIN2 promoter is dependent on BCL9/9L, β-catenin and also on TCF/LEF. These experiments, we believe, strongly support our model of TBX3 recruitment via the BCL9/β-catenin axis.

The reviewers also suggest that a proximality labeling experiment which can capture weaker interactions than co-IP could be done in Wnt-active versus inactive contexts to potentially confirm TBX3-Bcl9 interaction and identify other cofactors.

While it is true that proximity labeling assays have the potential to uncover novel dynamic and fleeting interactions, at this stage we consider that this goes beyond the scope of the present manuscript, for which we preferred to dedicate our efforts in reinforcing our understanding of the BCL9/TBX3 interaction. To this aim, we present a new set of compelling proximity ligation assays that show the cytological association between BCL9 and TBX3. Importantly, when quantified this association is comparable to that between BCL9 and PYGO (Figure 2—figure supplement 2). Moreover, please consider that the new ChIP-qPCR experiments described above, are all performed in both Wnt “ON” and “OFF” conditions, and they clearly support the simultaneous association of BCL9 and TBX3 on regulatory regions -hence proximity- in both these states. On the other hand, proximity labeling experiments are a very valuable suggestion, and performing them constitutes part of our future experimental plans.

The reviewers and editors point out that the authors should eliminate the term "enhanceosome" in the title and anywhere else. "Enhanceosome" has a very specific meaning in the transcription field, namely a highly structured physical entity composed of multiple transcription factors interacting with a complex and specific arrangement of target sites.

We agree with this concern. We have substituted the term “enhanceosome” with “Wnt/β-catenin transcriptional complex” throughout the text.

A reviewer's comment on phenotypes in seen Pygo1/2-KO and CTRL in regard to the length of villus and depth of crypts as well as number of proliferating cells. Therefore, in the absence of more detailed morphologic and quantitative histologic information, the description of the figure should be changed from "Intestinal architecture was normal" to "Not overt phenotypic defects".

We have removed the statement “Intestinal architecture was normal” and refer to the more cautious sentence “*Pygo1/2* deletion does not lead to any obvious histological or functional defect” (Figure 1B legend). Importantly also, we have added a careful quantification of the number of proliferative (Ki67^+^) cells in each genotype, taking into account several image fields and different mice. This quantification is normalized based on crypt length, and also supports the absence of obvious defects in the intestinal epithelium upon *Pygo1/2* or *Bcl9/9l* genetic depletion. This new analysis is shown in Figure 1—figure supplement 1D-E).

Reviewer #1:This manuscript builds upon the interesting observation that Bcl9/Bcl9L mutants have a more severe intestinal and limb defect than Pygo1/2 double mutants. This is interesting because the Bcl9s and Pygos are often thought to act in a complex to regulate Wnt targets. Using Bcl9 as bait, the authors then identify TBX3 as an interaction partner. TBX3 has limb deformities that might be similar to Bcl9/9L limb conditional knockouts. Using ChIP-seq of embryonic mouse limbs, they show an impressive co-localization of TBX3 and Bcl9 at over 1400 locations (2/3s of the total TBX3 peaks). Finally, they attempt to support a model where TBX3 is recruited to Wnt target gene enhancers through interaction with Bcl9/9L.The strengths of the manuscript are the convincing data demonstrating that Bcl9/9L mutants have more dramatic phenotypes than Pygo1/2. The ChIP-seq with Bcl9 and TBX3 is also very interesting. But the data functionally linking TBX3 to Bcl9 on Wnt targets is very weak and don't adequately support the molecular model that they present.

We appreciate the careful summary of our work, and the emphasis on the importance of the genetic observations as well as on the BCL9 and TBX3 ChIP-seq, which we also consider central to our finding. We also agree that the former version of our manuscript provided only weak evidence functionally linking TBX3 to BCL9 on target genes. As we will outline below, we present here a series of new ChIP-qPCR experiments strongly supporting the notion that physical occupancy of TBX3 on Wnt Responsive Elements (WREs), as well as the contribution of TBX3 to the transcription of these targets, strictly depends on the presence of β-catenin and its co-factors BCL9 and TCF/LEF.

1) It's not clear how similar that limb defects in Bcl9/9L mutants are compared to the TBX3 mutant limbs described in Frank et al., 2013. I disagree that they are "strikingly similar". Without more direct experimental observation, this connection needs to be related with more careful language.

We agree that we cannot be certain of the extent to which the defects caused by loss of *Bcl9/9l* recapitulate those induced by mutations in Tbx3, as this would require a fine anatomical and histological comparative analysis. As this however goes beyond our current scope, we decided to change the strength of our statement with a more cautious description. We now say: “The in vivo deletion of the HD1 domain (in *Bcl9/9l*-∆HD1 embryos) leads to severe forelimb malformations, while *Pygo1/2*-KO embryonic forelimbs are unaffected (Figure 2F)(also see Schwab et al., 2007). Limb development, thus, represents another context where BCL9/9L appear to act independently of PYGO. Of note, TBX3 plays a fundamental role in the development of this structure (Frank et al., 2013).” This change leaves unaffected the validity of our conclusion and prevents us venturing – as the reviewer suggested – into undocumented comparisons.

2) The data in Figure 4A,B does not address the model favored by the authors, i.e., that Bcl9/9L recruits TBX3 to enhancers. Rather it shows that genes with nearby enhancers bound by Bcl9 and TBX3 have decreased expression in Bcl9/9L mutants. This speaks to a requirement for Bcl9/9L, but doesn't address its relationship to TBX3.

We agree with the reviewer and have now removed the emphasis previously put in the description of this experiment. We have decided to move the presentation of this experiment “upstream” in the narrative, closer to the ChIP-seq experiment to which the comparison refers, clearly specifying the correlative nature of the data presented by saying: “Despite being of correlative nature, this analysis supports a model in which BCL9/9L and TBX3 cooperate to the activation of target genes”. The experiment is now presented in Figure 3G-H.

We still believe that this analysis provides support for a cooperation of BCL9 and an unknown factor – and the evidence points to TBX3 – for the transcription of Wnt target genes in vivo. We wish to point out that this RNA-seq analysis is executed with embryos carrying a trans-heterozygous allelic combination in which one allele has a deletion in the PYGO-binding domain (HD1) while the other lacks the β-catenin-binding domain (HD2) (*Bcl9*^ΔHD1/ΔHD2^; *Bcl9l*^ΔHD1/ΔHD2^, *Bcl9/9l*-Δ1/Δ2). In these embryos, the resulting BCL9 proteins can form BCL9/β-catenin or BCL9/PYGO complexes, but not the full tripartite β-catenin/BCL9/PYGO. Hence, *Bcl9/9l*-Δ1/Δ2 embryos do not have a “simple” loss-of-function of *Bcl9/9l*, and in these mice BCL9/9L retain both the HD1 and the HD2 binding capacities. Therefore, using the *Bcl9/9l*-Δ1/Δ2 animals is a way of testing the in vivo consequences of abrogating the tripartite complex mediated by the two interacting surfaces of BCL9/9L. As we show that *Pygo1/2* loss does not cause malformed forelimbs, the gene expression changes must be due to whatever other factor fails to be tethered by BCL9/9L onto the β-catenin transcriptional complex.

To clarify this, we have now added a better description of the genetic details relevant for this experiment in the manuscript.

3) A direct test of their model would be to see whether loss of Bcl9/9L affects TBX3 binding to enhancers. The authors have the reagents to do this key experiment.

We now present an entire new set of experiments in which we tested the requirement of BCL9/9L for the physical association of TBX3 on the Wnt Responsive Elements (WRE). We made use of a HEK293T cell clone devoid of BCL9/9L (kindly gifted by Mariann Bienz) and measured the enrichment of TBX3 association to the *AXIN2* promoter both in a “wild-type” as well as in a BCL9/9L-knockout (DB9/9L) context. We found that the association of TBX3 with the AXIN2 promoter in both “ON” and “OFF” states was dramatically reduced down to background levels upon loss of BCL9/9L (see Figure 4B and C).

We consider this experiment as a very important addition, as it shows that BCL9/9L are likely responsible for tethering TBX3 onto WREs (a tentative model that we represent in the right panel). Importantly, it also implies that BCL9 and TBX3 co-occupy (i.e. they sit simultaneously on the same position within the same cell).

While we recognize that this experiment could have been ideally carried out in an in vivo context (such as in *Bcl9/9l* mutant mouse forelimbs), we wish to point out that we only have been successful in performing BCL9 and TBX3 ChIP-seq when using ca. 500 forelimbs (130/IP sample). The genetic combination of the *Bcl9* and *Bcl9l* alleles we generated allows us to obtain a double homozygous embryo in a 1/16 ratio, which would render the collection of ca. 130 correct embryos impracticable. Additionally, the ChIP-qPCR setup in HEK293T allowed us also to test the requirement of β-catenin and TCF/LEF for TBX3 physical association in the same cellular context, by exploiting the D4TCF and Db-CAT cell clones used in this study (see Figure 4B and C, Doumpas et al., 2019). We could in fact show that also β-catenin and TCF/LEF are required for TBX3 binding on WREs (Figure 4B). While this was expected, as BCL9 requires β-catenin/TCF to sit on its targets, these experiments strongly corroborate the notion that the assembly of a Wnt/β-catenin dependent transcriptional complex is a prerequisite to allow TBX3 physical access to the WREs.

4) The overexpression data with TBX3 is unconvincing. At the concentration used, TBX3 represses Wnt targets in the absence of signaling. It's impossible to tell whether this is a direct or indirect affect (note the TBX3 ChIP data in Figure 4D indicates TBX3 is bound to the Axin2 promoter in the absence of signaling). The very small effect of TBX3 on TopFlash or Axin2 and Nkd1 is only "significant" when this repression is taken into account. This data is not acceptable evidence that TBX3 potentiates the expression of Wnt targets.

TBX3 has been previously described to act as a transcriptional repressor (Carlson et al., 2001). We consistently observe a repression effect when overexpressing TBX3, on both TOPFLASH (Figure 4A) as well as endogenous targets (Figure 4—figure supplement 1). The repression switches to an activation effect when CHIR is added to the culture (see the use of 1mM CHIR on the reporter in Figure 4A, and the considerable increase in the fold enrichment in the activation of *AXIN2* and *NKD1*, Figure 4—figure supplement 1). Please note that the logarithmic scale on the y-axis of Figure 4A, left panel, might mask this effect. At 1mM CHIR, on average in our hands, while HEK293T cells activate the reporter 10-fold, while TBX3-expressing cells reach values of ca. 80/100, close to saturating level in this setup. We have added a brief note within the figure caption to mark the difference in scale representation.

We agree with the reviewer that the mechanism by which this happens is not clear, and that this is the only evidence we provide that TBX3 potentiates the expression of Wnt target genes. Therefore, we decided to add a statement of caution in the main text, and now suggest that: “While our experiments show that TBX3 can influence the expression of Wnt target genes, the mechanisms by which this occurs remain to be elucidated”. Overall however, we wish to point out that this does not invalidate the main conclusion of our study, consisting in the physical participation of TBX3 to the Wnt/β-catenin dependent transcriptional complex. First of all, the physical occupancy of TBX3 also in the Wnt OFF state is in line with recent studies indicating that most of the β-catenin co-factors (including BCL9) assemble before the activation of the pathway and the “arrival” of β-catenin (van Tienen et al., 2017). Moreover, the new ChIP experiments in a BCL9/9L-KO context that we present in the revised version of the manuscript strongly reinforce our model. Hence, while understanding the exact mode of action of TBX3 at these loci, as well as the quantitative effect on their transcriptional output, remain an important question that we wish to address in a more mechanistic follow-up study, we now think that the experiments presented are convincing in suggesting that the effect of TBX3 on Wnt targets is direct and it is mediated by its TCF/β-catenin/BCL9-dependent recruitment on WREs.

5) The zebrafish metastasis data is interesting, but it's unclear whether the ability to spread from sight of injection is dependent on Wnt signaling. So it's impossible to interpret whether it supports that authors' model.

We agree that this experiment does not formally prove that the TBX3-dependent enhancement of the number of metastases we observed is strictly dependent on Wnt signaling. We decided to render this explicit, by writing that: “While this experiment does not allow to exclude that TBX3 might also act independently of BCL9/β-catenin in this context, it shows that increased expression of TBX3 enhances proliferation and migratory capability of human CRC cells bearing constitutively active Wnt signalling”. On the other hand, we now also show a new analysis indicating that TBX3 overexpressing HCT116 cells maintain high expression of TBX3 throughout the course of 3 days (during which these human cells migrate through fish tissue) and this is accompanied by enhanced Wnt signalling (as measured by increased AXIN2 expression). This was made possible by measuring human-specific gene expression after total mRNA extraction from pools (2 independent experiments of N=10 fish each) of zebrafish embryos. This analysis is now shown in the new panel in Figure 4G

We interpret this as evidence that TBX3 can induce increased number of metastases in this model, and that this is associated with simultaneous enhancement of the Wnt/β-catenin-dependent transcription.

6) They should test whether depletion of TBX3 affects Wnt regulation of Axin2 (their ChIP data in Figure 4D suggests Axin2 might be a direct TBX3 target). They should also test whether the ChIP signal at the Axin2 enhancer and promoter are dependent on β-catenin and TCFs (with the CRISPR lines they use in Figure 4C). These experiments directly test their model with reagents that they already have in hand.

As described before in response to comment 3, we now present a new set of ChIP experiments clearly showing that TBX3 recruitment on the *AXIN2* promoter is dependent on the presence of not only BCL9/9L, but also on β-catenin and TCF/LEF (Figure 4B and C). We are grateful for this comment, as it provided the impetus for performing this series of experiments that we now consider among the most prominent evidence in favor of our hypothesis – that the BCL9/β-catenin/TCF complex recruits TBX3 on Wnt-dependent regulatory regions.

Concerning the other point, if depletion of TBX3 affects the regulation of *AXIN2*: at this stage, we cannot provide any conclusive evidence. In the course of the last several months, we attempted many strategies to downregulate or knockout *TBX3* from a number of cellular models, with however negative results. So far, our observations are suggestive of a strict requirement for TBX3 for the viability of the cellular models at our disposal. We wish to point out that, while this is a very interesting observation per se, and we will explore it in more depth, it is also reminiscent of the identification of TBX5 as a necessary factor required for the viability of β-catenin-dependent cells (Rosenbluh et al., 2012). Whether TBX3 and TBX5 act in concert with β-catenin in a similar fashion is an interesting research question worth of future investigation.

Reviewer #2:Aberrant activation of the canonical Wnt/β-catenin pathway drives the formation of various human cancers and has emerged as a promising, yet challenging, target for cancer therapy. BCL9/9L and PYGO1/2 are transcriptional co-activator's of β-catenin and it is believed that their concerted action is required for efficient activation of Wnt/β-catenin-target gene expression and tumor phenotype. In this manuscript, by using various strains of engineered mice, Zimmerli et al. provide genetic evidence indicating that BCL9/9L function does not entirely depend on PYGO1/2. They also showed that during forelimb development, BCL9/9L have a PYGO1/2 independent function, and identified TBX3 as a candidate tissue-specific transcriptional factor that functional interact with BCL9/9L in mediating Wnt/β-catenin transcriptional output.This is a clear, well written manuscript, with experimentally supported conclusions, which can have important basic, translational and clinical implications.1) The histologic characterization of CTRL and Pygo1/2-KO intestine and colon should be more systematic, detailed and quantified using digital pathology (e.g. Halo Analysis platform). From the representative pictures provided in Figure 1B and Figure 1—figure supplement 1C it seems that the villi and crypts are slightly shorter and the number of Ki-67 positive cells is reduced in Pygo1/2-KO mice.

Our former conclusions were based on (i) the absence of any phenotypic consequence upon *Pygo1/2* conditional deletion in the intestinal epithelium, and (ii) the lack of any obvious histological malformation. But we agree that a more reliable quantification of the parameters suggested is beneficial to support our conclusions. We have now measured both the number of proliferating cells in our mutants compared to control animals, and the average crypts depth in the colon and villus length in the small intestine. The Ki67+ fraction was calculated by normalizing the length of the proliferative compartment by the relative crypt villus length, and 10-15 crypts were scored for each biological replicate considering at least 3 different mice per genotype. The new data are now represented in Figure 1—figure supplement 1D-E. We are grateful for this suggestion, as these data make us more confident that, similarly to BCL9 proteins, also PYGO factors are largely dispensable for the homeostasis of the intestinal epithelium.

2) In Figure 2E, because the epitope recognized by the BCL9 antibody could be masked during potential BCL9/TBX3 direct interaction, this reviewer is wondering whether the IP should be done with two anti-BCL9 antibodies, and a reverse IP using anti TBX3 antibodies should be also used.

We have attempted several strategies to detect the interaction between BCL9 and TBX3 in a Co-IP assay, however with non-conclusive results. Our trials included the use of an anti-BCL9 or an anti-GFP antibody to pull down a BCL9-GFP fusion protein, in addition to the anti-HA antibody (against the BCL9-HA) used in this study, and the reciprocal anti-FLAG antibody when expressing a TBX3-FLAG protein. Hence it is unlikely that the epitope becomes masked by a specific antibody. Rather, it is plausible that the interaction between BCL9 and TBX3 is of dynamic nature and occurs in a rapid succession with other events. Alternatively, TBX3 and BCL9 could require other factors, such as tissue-specific additional players or the proximity to the DNA. Investigating these possibilities is among our experimental priorities for follow-up mechanistic studies. Nevertheless, we believe that throughout the study we present strong evidence in favor of their physical vicinity in a living cell. New evidence derives from the new PLA assay presented (Figure 2—figure supplement 2), showing the proximity of BCL9 and TBX3 within the nucleus of transfected cells. The second is the compelling observation – also obtained in the context of this revision – that mutations in *BCL9/9L* abrogate TBX3 binding on the *AXIN2* promoter, as seen by ChIP of TBX3-FLAG in HEK293T carrying mutations in *BCL9* and *BCL9L* (new Figure 4B-C). This not only suggests co-occupancy of regulatory regions – hence physical and functional proximity -, but also that BCL9 is required for recruitment of TBX3 on WREs.

3) In Figure 3F, it would be important to know what happens with the co-occupancy of BCL9 and TBX3 within the Wnt responsive element of Lgr5.

This is an interesting suggestion. We looked at the *Lgr5* locus across our experimental replicates (see Author response image 1). The genomic tracks reveal two possible binding regions, also in this case presenting precisely overlapping positioning of BCL9 and TBX3 (both highlighted by transparent light blue areas). One potential binding region is in the proximity of the promoter (on the right of the image), and another close to the second exon (at the center). The peak on exon 2 is called for both TBX3 samples but for only one of the BCL9 samples, and this prevented the peak being listed among the set of common TBX3/BCL9 peaks in our conservative approach. The peak close to the promoter is not called in any of the samples, likely because the signal at this position is not high enough in comparison to the surrounding noise. Nevertheless, this observation might suggest that, even in some limb cells, *Lgr5* is a TBX3>Wnt/BCL9 target gene. It is relevant to keep in mind that these are forelimb cells, while the relevance of *Lgr5* as a target gene is clearly established for the intestinal epithelium. This, we think, might be an additional common feature between these two tissues.

**Author response image 1. sa2fig1:** 

4) In legend of Figure 4, the label (D) before (C) should be eliminated or changed according to other labels.

We have corrected this mistake.

Reviewer #3:[…]Some follow up questions include the extent of TBX3 involvement in activation of Wnt target genes in other contexts, and whether that depends on Bcl9. Does increased STF induction or tumorigenesis upon TBX3 overexpression require Bcl9, for instance? Does loss of TBX3 itself affect Wnt target activation?

There are a number of follow up questions that our study left open, such as the important one concerning the reciprocal requirement between TBX3 and the Wnt co-factors BCL9, β-catenin and TCF/LEF raised by this and other reviewers. We now present a novel set of ChIP experiments which revealed that, in the absence of BCL9/9L, or of β-catenin and TCF/LEF, TBX3 fails in physically associating to the AXIN2 promoter (new panel in Figure 4B and C – see also response to reviewer 1). As a consequence, modulation of TBX3 expression does not affect STF transcription in the absence of the main Wnt-dependent transcriptional transducers β-catenin and TCF/LEF (Figure 4A). This is an important addition to our study, as it clearly shows the requirement of the Wnt transcriptional complex for the binding of TBX3 on Wnt Responsive Elements (WREs), and its role in transcriptional regulation at these loci. It remains to be determined what the fraction in, on a genome-wide scale, of the TBX3 target loci for which the axis BCL9/β-catenin/TCF is necessary for direct binding. We consider this an interesting new question for future follow up studies. Ideally, this experiment should be carried out in an in vivo context, such as in *Bcl9/9l* mutant mouse forelimbs. At this stage however, we have been successful in performing BCL9 and TBX3 ChIP-seq when using ca. 500 forelimbs (130/sample). The genetic combination of the *Bcl9* and *Bcl9l* alleles we generated allows us to obtain a double homozygous embryo in a 1/16 ratio among littermates; this renders the collection of ca. 130 *Bcl9* and *Bcl9l* double mutant embryos impracticable at the moment.

Concerning the important question if a loss of TBX3 affects the regulation of target genes: while we consider this a very relevant aspect of our investigation, at this stage we cannot provide any evidence describing the consequence of TBX3 loss. In the course of the last months, we attempted many strategies to downregulate or knockout *TBX3* from a number of cellular models, with however negative results. So far, our observations are suggestive of a requirement of TBX3 for viability of the cellular models in our hands. While this is an interesting observation per se, it is currently preventing us to fully grasp the consequences of TBX3 loss-of-function in the context of Wnt signaling. Also, our observation is reminiscent of the identification of TBX5 as a necessary factor required for viability of β-catenin-dependent cell (Rosenbluh et al., 2012). Whether TBX3 and TBX5 act in concert with β-catenin in a similar fashion is a powerful research question worth future investigation. We have started to explore this possibility more in depth, and our plans include the use of mouse models and the induction of tumors in vivo, benefiting of conditional *Tbx3* alleles (Frank et al., 2013). But the time necessary for the execution of these experiments (and the mouse crosses) imposes us to consider them for a subsequent report.

It would also be interesting to further probe TBX3-Bcl9 or Tbx3-β-catenin interaction.

As described in response to a comment from other reviewers, we have attempted to detect the interaction between BCL9 and TBX3 in a Co-IP assay in many ways, but our results are not conclusive. For example, we used anti-BCL9 or anti-GFP antibodies to pull down endogenous BCL9 or a BCL9-GFP fusion protein, in addition to the anti-HA described in the manuscript. We have also assessed the reciprocal IP, using an anti-FLAG against a TBX3-FLAG protein and probing the presence of BCL9 in the IP reaction. In several of these experiments, TBX3 appeared “sticky”, but its affinity for BCL9 was comparable to that for GFP alone, and the use of higher salt concentration abrogated both these interactions. We speculate that, in vivo, the interaction between BCL9 and TBX3 might be dynamic, perhaps occurring in rapid succession with other events during transcriptional regulation. Alternatively, the interaction itself could require the presence of other co-factors (present in limbs but not in other cells) or be even conditional to the proximity of DNA. Investigating these possibilities is among our experimental priorities in follow up studies. On the other hand, we already present strong evidence in favor of their physical vicinity and functional interplay. The first evidence derives from the quantification of new PLA assays (Figure 2—figure supplement 2 – see also the response to reviewer 1) showing that the proximity of transfected BCL9 and TBX3 occurs within the nucleus at levels comparable to that between BCL9 and PYGO2. The second, even more compelling, and also obtained in the context of this revision, is the new finding that mutations in *BCL9/9L* abrogate TBX3 binding on *AXIN2* promoter, as seen by ChIP-qPCR assays (Figure 4B, C). This, in our opinion, now constitutes one of the main evidence presented in our study, as it suggests physical proximity on the same molecule of DNA (i.e. co-occupancy), but also that BCL9 is necessary for the recruitment of TBX3 on WREs.

An RNA-seq experiment with Bcl9 missing HD1 but intact HD2 domain, or TBX3 ChIP-seq in the context of Bcl9 HD1 deletion, would further support functional interaction of Bcl9 HD1 and TBX3 in Wnt target activation.

During the execution of the current study we have chosen to focus on the trans-heterozygous allelic combination in which embryos carry deletions in the PYGO-binding domain (HD1) on one allele and deletion of the β-catenin-binding domain (HD2) on the other (*Bcl9*^ΔHD1/ΔHD2^; *Bcl9l*^ΔHD1/ΔHD2^, *Bcl9/9l*-Δ1/Δ2). In these embryos, the resulting BCL9 proteins can form BCL9/β-catenin or BCL9/PYGO complexes, but not the full tripartite β-catenin/BCL9/PYGO. Hence, *Bcl9/9l*-Δ1/Δ2 embryos do not carry a “simple” loss-of-function of *Bcl9/9l*, because in these mice BCL9/9L retain both the HD1 and the HD2 domains. As BCL9/9L produced by one allele can use the HD1 for interacting (but not the HD2) and the other still possesses the HD2 (but not the HD1), this allelic combination is a specific way of testing the consequences of abrogating the tripartite complex mediated by the two interacting surfaces of BCL9/9L. Therefore, as we show that *Pygo1/2* loss does not cause malformed forelimbs, the gene expression chances must be due to whatever other factor fails to be tethered by BCL9/9L onto the β-catenin transcriptional complex. We add a clearer description of the genetic details relevant for this experiment in more depth in the manuscript. For this reason, we think that the allelic combination that we used is a more informative experiment for the current purpose of this study.

However, we agree that the HD1 deletion – leaving the HD2 intact – would also be a way of abrogating the interaction with the new interactors (e.g. TBX3). This is exemplified by the observation that the homozygous deletion of HD1 in *Bcl9/9l* also causes forelimbs malformations (Figure 2F). An RNA-seq analysis of forelimbs with only the HD1 mutated would be of certain interest, as it would lead to a likely broader set of consequences many of which might not depend on the action of BCL9/9L in the context of Wnt signalling. We have previously shown that BCL9/9L, especially via the HD1, entertain Wnt/β-catenin independent roles (Cantù et al., 2014; Cantù et al., 2017). However, we could not consider this experiment a priority for the present study. This is also due to the low probability of the breeding outcomes (each double mutant embryo occurs with a probability of 1/16), that would have made it very difficult to perform both types of breedings in parallel, or the second one (to obtain mutation only in HD1) in the limited time for this revision.

Additionally, a proximality labeling experiment that can capture weaker interactions than co-IP could be done in Wnt-active versus inactive contexts to potentially confirm TBX3-Bcl9 interaction and identify other cofactors.

While it is true that proximity labeling assays have the potential to uncover novel dynamic and fleeting interactions, at this stage we considered this as a scope that goes beyond that of the present manuscript. During this revision, we dedicated our efforts in reinforcing our understanding of the BCL9/TBX3 interaction rather than discovering new potential candidates. To this aim, we present a new set of compelling proximity ligation assays that show a cytological association between BCL9 and TBX3 that, when quantified, is comparable to that between BCL9 and PYGO (Figure 2—figure supplement 2). Moreover, please consider that the new ChIP-qPCR experiments described above, are all performed in both Wnt “ON” and “OFF” conditions, and they clearly support the simultaneous association of BCL9 and TBX3 on regulatory regions in both these states.

However, we consider the use of proximality labeling assays as a very valuable suggestion. This would imply the generation of APEX2 or BirA fusion proteins for BCL9 and TBX3, and we will consider this as an important part of our future experimental plans.